# Real-time resolution studies of the regulation of pyruvate-dependent lactate metabolism by hexokinases in single cells

Scott John[1], Guillaume Calmettes[1], Shili Xu[2], Bernard Ribalet [3] *

1 Department of Medicine (Division of Cardiology), David Geffen School of Medicine at UCLA, Los Angeles, CA, United States of America, 2 California NanoSystems Institute (CNSI) 2151, David Geffen School of Medicine at UCLA, Los Angeles, CA, United States of America, 3 Department of Physiology, David Geffen School of Medicine at UCLA, Los Angeles, CA, United States of America

* bribalet@mednet.ucla.edu

## Abstract

Lactate is a mitochondrial substrate for many tissues including neuron, muscle, skeletal and cardiac, as well as many cancer cells, however little is known about the processes that regulate its utilization in mitochondria. Based on the close association of Hexokinases (HK) with mitochondria, and the known cardio-protective role of HK in cardiac muscle, we have investigated the regulation of lactate and pyruvate metabolism by hexokinases (HKs), utilizing wild-type HEK293 cells and HEK293 cells in which the endogenous HKI and/or HKII have been knocked down to enable overexpression of wild type and mutant HKs. To assess the real-time changes in intracellular lactate levels the cells were transfected with a lactate specific FRET probe. In the HKI/HKII double knockdown cells, addition of extracellular pyruvate caused a large and sustained decrease in lactate. This decrease was rapidly reversed upon inhibition of the malate aspartate shuttle by aminooxyacetate, or inhibition of mitochondrial oxidative respiration by NaCN. These results suggest that in the absence of HKs, pyruvate-dependent activation of the TCA cycle together with the malate aspartate shuttle facilitates lactate transformation into pyruvate and its utilization by mitochondria. With replacement by overexpression of HKI or HKII the cellular response to pyruvate and NaCN was modified. With either hexokinase present, both the decrease in lactate due to the addition of pyruvate and the increase following addition of NaCN were either transient or suppressed altogether. Blockage of the pentose phosphate pathway with the inhibitor 6-aminonicotinamide (6-AN), abolished the effects of HK replacement. These results suggest that blocking of the malate aspartate shuttle by HK may involve activation of the pentose phosphate pathway and increased NADPH production.

## Introduction

There is now evidence that lactate is a mitochondrial substrate for many tissues including neuron, skeletal, and cardiac muscles [1–5]. However, little is known about the regulation of lactate utilization by mitochondria, although there is a broad base of knowledge regarding the

**Data Availability Statement:** All relevant data are within the paper and its Supporting Information files.

**Funding:** The author(s) received no specific funding for this work.

**Competing interests:** The authors have declared that no competing interests exist.

metabolism of pyruvate and fatty acids in these organelles. For instance, it is known that upon feeding and under the influence of insulin, there is a switch from fatty acid oxidation to glucose oxidation, which is often referred to as the Randle cycle [6–9]. There is also evidence that binding of HKs to the mitochondria outer membrane (OMM) plays a role in this process by increasing the coupling between glycolysis and complete mitochondrial glucose oxidation, and by blocking fatty acid oxidation [10–12]. It should be noted that few studies mitigate this report, and suggest that HKs uncouple glycolysis and glucose oxidation under some experimental conditions [13,14].

## Modulation of lactate utilization by mitochondria

For most of the 20th century, lactate was considered a dead-end waste product of glycolysis. Becoming elevated during hypoxia (anaerobic metabolism), it was assumed to be responsible for muscle fatigue [15], as well as being a key factor in acidosis-induced tissue damage [16,17]. Since the early 1980s, a different aspect of lactate metabolism has emerged, whereby its transformation into pyruvate may fuel the TCA cycle [1–3]. However, lactate is not permeable through the mitochondria inner membrane, and therefore cannot be used directly by mitochondria. To account for this limitation, a model has been proposed whereby a lactate dehydrogenase enzyme associated with the mitochondrial intermembrane space (mLDH) transforms lactate into pyruvate, which is then transported into the mitochondrial matrix [1–3,18–24]. It has been suggested that the transformation of lactate into pyruvate by mLDH in the intermembrane compartment is dependent upon the production of $NAD^+$ by the malate aspartate shuttle (MAS) [25–27]. Within this scheme, the activity of the mLDH is coupled with the activity of the mitochondrial pyruvate dehydrogenase (PDH) which creates a sink for pyruvate and thereby allows flux of the substrate into the mitochondrial matrix [25,26]. The role of MAS in the transformation of lactate into pyruvate has been demonstrated using Aminooxyacetic acid (AOA). AOA has been broadly used as an inhibitor of two aminotransferases, 4-aminobutyrate aminotransferase (GABA-T) [28] and aspartate aminotransferase. Aspartate aminotransferase is a PLP-dependent enzyme, and an essential part of the malate-aspartate shuttle. Thus, inhibition of the malate-aspartate shuttle by AOA has been shown to prevent the reoxidation of cytosolic NADH by the mitochondria in tissues such as nerve terminals, vascular smooth muscle and the heart. In nerve terminals, AOA prevents mitochondria from utilizing pyruvate generated from glycolysis, leading to a bioenergetic state similar to that of hypoglycemia [29]. It also blocks the use of lactate to generate energy in synaptic terminals [2]. In heart inhibition of reoxidation of cytosolic NADH by AOA causes lactic acidosis, and ultimately impairment of myocardial function [30,31].

## Modulation of lactate utilization by mitochondria by hexokinases I and II

Evidence has been provided in favor of MAS-produced $NAD^+$ regulating the transformation of lactate into pyruvate, but little is known about the mechanism(s) that controls this process and how it may vary among different tissues. We propose that HKs are part of this regulatory process.

Most cell types express the two hexokinases I and II (HKI and HKII), but HKI is more abundant in fast proliferating cells and in neurons [32,33], whereas HKII is highly expressed in cells like those in cardiac and skeletal muscle [34]. Many cell lines express a combination of both HKs. Accordingly, HEK cells, the cells used in our study, express both HKI and HKII, with HKII being higher than HKI. HKI and HKII have different properties. HKII has a higher catalytic activity and binds to the OMM in a highly dynamic manner, with kinases such as the serine/threonine kinase Akt strengthening its interaction with the OMM and G6P (the HK

catalytic product) causing its dissociation. In contrast, HKI strongly binds to OMM and cannot be displaced by most metabolic perturbations [11,35–38].

We have carried out experiments with the specific FRET sensor "Laconic" to monitor intracellular lactate in real time in HEK cells. For our purpose HKI and HKII could be knocked-down either individually or together in a doxycycline-dependent manner. We have used the HK knocked-down HEK cells to reintroduce the HKs one at a time, thereby investigating their individual roles in regulating the metabolism of pyruvate and lactate. We have used these cells for our experiments instead of wild-type HEK cells, because the effects of pyruvate and NaCN were inconsistent from experiment to experiment in wild-type HEK cells, rendering these cells less than satisfactory for our work (see S2 Fig). Altogether, our data suggests that the malate aspartate shuttle (MAS) plays a role in lactate utilization by mitochondria via transformation of lactate into pyruvate in the mitochondria's intermembrane space and this process is inhibited by hexokinases (HKs) binding to the OMM.

## Materials and methods

### Solutions and experimental techniques

The bath solution for cell imaging consisted of (in mM) 140 NaCl, 5 KCl, 1.1 $MgCl_2$, 2.5 $CaCl_2$, 10 HEPES, with the pH adjusted to 7.2 with NaOH. $N$-methyl-$d$-glucamine was added to maintain the solutions' osmolarity when glucose was lowered. For most experiments 5 mM glucose was added to the bath solutions (with or without HK expression), so that HK would be active when expressed. Solutions were perfused directly over the cells using a gravity-fed eight-way perfusion device (Warner Instruments, Hamden, CT, USA) with electrically controlled solenoids (The Lee Company, Westbrook, CT, USA). Input and output of solution volumes to the recording chamber (glass bottomed Petri dish) were equilibrated to maintain constant flow rates and pressures within the recording chamber. Aminooxyacetic acid (AOA), Sodium Cyanide (NaCN) and 6-Aminonicotinamide (6-AN) were purchased from Sigma-Aldrich. The inhibitor of ME1 (ME1*) was purchased from (ProbeChem biochemicals, shanghai).

### HKI and HKII knock down and overexpression in HEK cells. Molecular biology and cell culture

Hexokinase wild-type and mutant constructs were subcloned into the pcDNA3.1 amp mammalian expression vector (Invitrogen), which utilizes the cytomegalovirus promoter. HEK293 cells were transfected with lipofectamine 2000 (Invitrogen). Cells were cultured in Dulbecco's modified Eagle's medium, high (25 mM) or low (5.5 mM) glucose medium supplemented with 10% ($v/v$) fetal bovine serum (FBS), penicillin (100 U/ml), streptomycin (100 U/ml), and 2 mM glutamine. Cells were divided twice a week by treatment with trypsin. To knock down HKs HEK293 cells were treated with 0.5μg/ml doxycycline (DOX) for 48 hrs, starting 24 hrs after plating. For (DOX)-inducible shRNA knockdown of HKs we followed the protocol first introduced by Krall et al. in 2016 to examine tumor cell growth inhibition in liver cancer cell lines (HK1−HK2+ and HK1+HK2+ H460). They showed in their study that shHK2 expression in the HK1−HK2+ H460 suppressed cell proliferation [39].

Briefly, the approach to DOX-inducible shRNA HK knockdown was as follows: 1) Five shRNA sequences proposed to target HK mRNA were indentified. shHK-1, shHK-2, shHK-3, and shHK-4 targeted the coding sequence of the HK mRNA. shHK-5 targeted the 3' untranslated region of the HK mRNA. 2) Sequencing of the shRNAs was performed by Laragen Inc (Culver City, CA). 3) HK knockdown efficacy of the five shHK2 sequences was tested in stable isogenic cells generated with DOX-inducible shRNAs integrated in the genomes using

lentivirus. The isogenic cells were cultured with or without DOX (25 ng/mL) for three days prior to analysis of HK1 and HK2 protein levels by Western blotting. HK1 (#2024), HK2 (#2867), and GAPDH (#5174) antibodies obtained from Cell Signaling Technology were used for Western blots. Results of an immunoblotting experiment illustrated in S6 Fig showed successful specific depletion of HK1/HK2 or both in our cells.

## FRET sensors

Most of the methods used to measure substrate such as lactate are based on enzymatic reactions and have limitations, since they are not able to detect changes in intracellular substrate levels non-invasively in real-time or with single cell resolution. To circumvent these limitations genetically-encoded reporters have been engineered to monitor with improved spatio-temporal resolution the transport and utilization of substrates, including lactate. Laconic is a genetically-encoded Forster Resonance Energy Transfer (FRET)-based lactate sensor that has been designed with the bacterial transcription factor LldR at its core. This region which forms the ligand-binding domain is flanked by a pair of fluorescent proteins, mTFP (equivalent to CFP) and Venus (equivalent to YFP). These two fluorescent proteins have overlapping emission and excitation spectra. Binding of the substrate to the sensor causes a conformational change that affects the relative distance and/or orientation between the two fluorescent proteins, increasing or decreasing FRET efficiency as a result (Fig 1A). In the case of the lactate FRET sensor called laconic, the response to binding of lactate is nearly linear between $10^{-5}$ and $10^{-2}$ M and is not affected by changes in the level of other substrates, such as glucose, pyruvate, glutamate, malate or oxaloacetate at cytosolic concentrations [40]. Furthermore, it has little sensitivity to changes in pH in the physiological range [40,41]. Complementary DNA (cDNA) for Laconic was purchased from Addgene (plasmid # 44238) and transfected into HEK 293 cells with lipofectamine 2000 (Invitrogen). Expression of Laconic was sufficiently high after 24 hrs to perform FRET measurements. In our system the probe was expressed in the cytoplasmic compartment, since it had no organelle-targeting sequences associated with it; it therefore reports "global" cytoplasmic lactate levels.

## FRET imaging

Images (16-bit) were acquired using a Nikon Eclipse TE300 microscope fitted with a ×60 (N.A. 1.4) oil immersion lens (Nikon) and equipped with a filter cube comprising a CFP bandpass excitation filter, 436/20b, together with a longpass dichroic mirror (Chroma Technology Corp, Rockingham, VT, USA). Light-emitting diodes (LEDs, Lumileds, San Jose, CA, USA) were used as light sources: one emitting at 455±20 nm (royal blue) and the other emitting at 505±15 nm (green/yellow). LEDs and camera exposure were controlled by MetaFluor Imaging 6.1 software (Molecular Devices, Sunnyvale, CA, USA).

Each experiment consisted of simultaneous and separate recordings from four to five cells. Regions of interest (ROIs) were drawn around each cell and FRET signals were integrated over these regions. Calculated FRET ratios were saved for each cell (Fig 1B). FRET ratios from 30 to 100 individual cells were analyzed for each experimental condition. For each experimental condition, recordings from "control cells" were carried out on the same day.

Ratiometric FRET measurements were performed by simultaneously monitoring CFP and YFP emissions of the sample when excited at the wavelengths for CFP (455±20 nm). The ratio between YFP and CFP emission was measured online in real time using MetaFluor Imaging software. For analysis, background light intensity was subtracted from the individual YFP and CFP emission. YFP and CFP images were acquired simultaneously using a Dual View image splitter (Optical Insights, Tucson, AZ, USA) equipped with a 505-nm long-pass dichroic filter

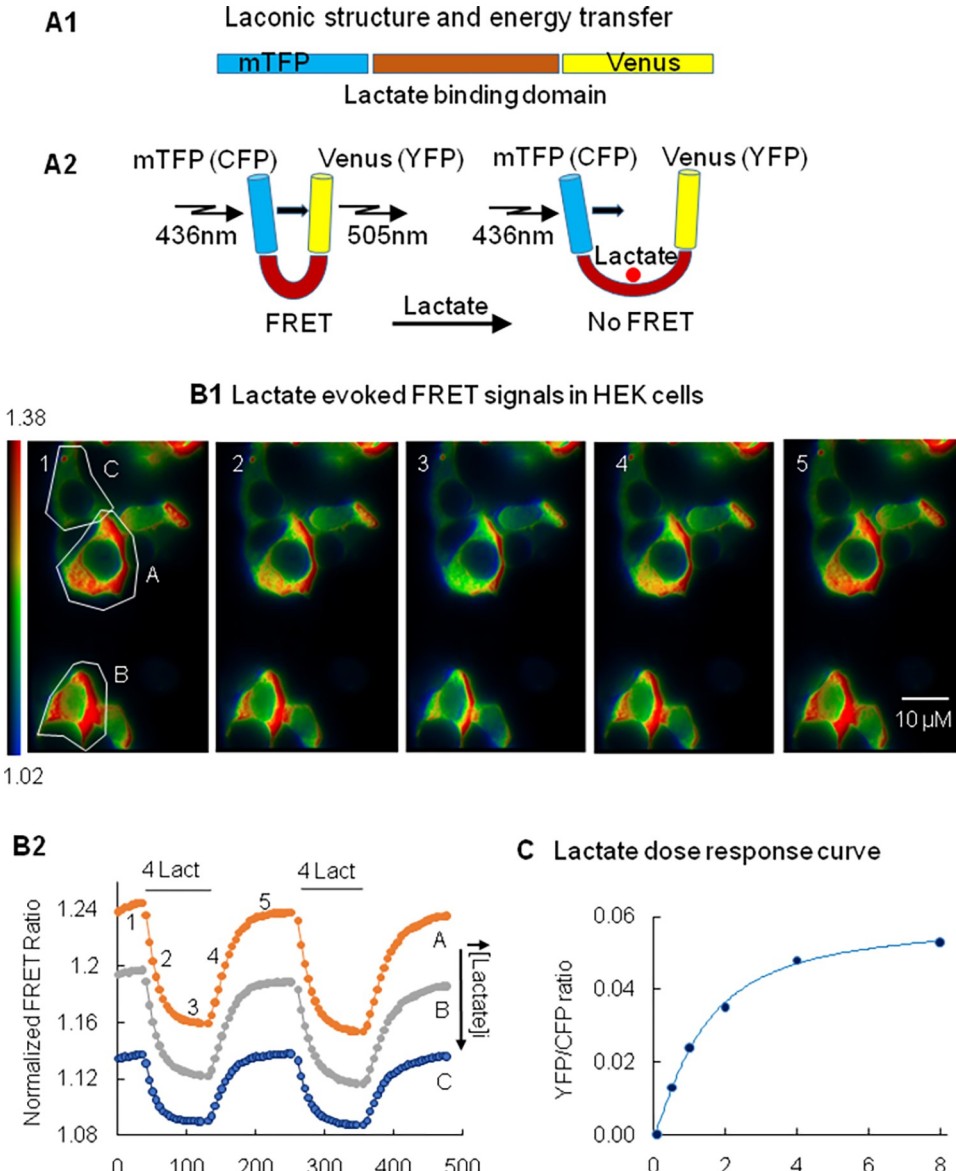

**Fig 1. Methodology.** Panel A1. Structure of the Laconic sensor. The sensor comprises three main regions that form the cytoplasmically expressed protein, the mTFP domain (CFP excitation), the bacterial transcription factor LldR, which binds lactate, and the Venus domain (YFP emission) [40]. Panel A2. Model of the changes in FRET evoked by lactate binding. In the absence of lactate the donor (CFP) and the acceptor (YFP) are within 1–50 nm of each other and there is significant fluorescent energy transfer (YFP/CFP ratio is high). Upon binding of lactate YFP and CFP move farther apart, YFP intensity decreases while CFP intensity increases. This leads to a decrease in the FRET ratio YFP/CFP. Panels B1 show images of the changes in FRET ratio evoked by the addition of 4mM lactate. The pseudo color images are shown for illustration purpose, but in most experiments, we only recorded the ratio values shown in panel B2 (not the raw images in panel B1). Every FRET experiment encompassed measurements from four to six individual cells at the same time in the same microscope field of view. A region of interest (ROI) was drawn around each cell and FRET ratios were recorded in real time over these ROIs. Panel B1 illustrates how addition of lactate, starting at image #1 and ending at image #3, causes a decrease in FRET ratio from 1.38 to 1.02 (see scale bar on the left of panel B1). Upon removal of lactate, starting at image #3 and ending at image #5, the FRET ratio recovered (increase in light intensity). The number of each image corresponds to the numbers on the trace A of panel B2. Panel B2 illustrates how the FRET ratio integrated over the ROI in each cell (shown in panel B1), with varying expression levels of laconic (A, B, C), changes upon addition and removal of lactate applied extracellularly. Here the ratio YFP/CFP is plotted as a function of time. Again, an upward trend of the FRET ratio corresponds to a decrease in intracellular lactate level and vice versa a downward trend indicates an increase in lactate level. These YFP/CFP ratio values were recorded and kept for further

analysis. In the experiments presented in this report, data from thirty to hundred cells, taken over separate experiments were used for statistical analysis. Even though the level of expression of the sensor laconic affected the absolute value of the FRET ratio (compare traces A, B, C) the time course of the changes in FRET ratio was similar. This observation suggests that the FRET sensor does not have a strong effect in "buffering" intracellular lactate. Panel C. Graph of the changes in FRET ratio plotted as a function of concentration and fitted with a Hill equation. The fits yield a Kd of 1.26 mM. This value obtained from the intracellular milieu is close to that reported by San Martin et al. in 2013 for in vitro experiments (830±160 μM) [40]. Because the FRET ratio values reported in our results are expressed as a percentage of the maximal change obtained at saturating concentration of lactate, these values may be used to get an estimated of the changes in concentration observed under the various experimental conditions. More details on this dose response curve is shown in S1 Fig.

to separate the CFP and YFP signals, coupled with a CFP emission filter (480/30), and a YFP emission filter (535/40). Acquisition of the CFP and YFP images, captured with a Cascade 512B digital camera (Photometrics, Tucson, AZ, USA), was carried out using the imaging software. Exposure times were optimized in each case but varied between 100–500 ms and were recorded at a constant rate for each cell between 0.2–0.33 Hz. Many experiments lasted more than 1 h, leading to a slow drift in the FRET ratio baseline in some cases. In those cases, the drift was corrected using either linear or exponential curve fitting, as appropriate.

## Statistical analysis

Most changes in FRET ratio due to various experimental perturbations were fitted with a single exponential, a combination of two exponentials or a combination of exponential and sigmoid functions (S1 Fig). The fitted amplitude of the FRET signals was then normalized and expressed as a percentage of control, measured with addition of 4mM lactate (Fig 2A). To test inhibitors (incubation overnight) or overexpression of HKs we used the same batch of cells (same passage number) for the test and control FRET experiments. Histograms with bin size of 10% were used to display the amplitude of the different phases under the various experimental conditions. Normality and homogeneity of variance were evaluated by Shapiro-Wilk test and F-test two-samples respectively. In the case of non-homogeneity (indicated in figure legends) a t-test two-samples assuming unequal variances was used to evaluate statistical significance. Otherwise, a t-test two-sample assuming equal variances was used. Mean ± SEM, standard deviation and P values are reported in the text and figure legends. A critical value for significance of $P < 0.05$ was used throughout the study. When statistical thresholds of 0.05, 0.005 or 0.0005 are used it is indicated in the figure legends.

## Results

### Evaluation of the mitochondrial metabolism of pyruvate and lactate using a FRET-based lactate sensor

Binding of hexokinases (HKs) to the outer membrane of mitochondria (OMM) is known to be a potent regulator of substrate utilization by mitochondria. Interaction of HKs with OMM downregulates fatty acid oxidation and upregulates the coupling between glycolysis and pyruvate oxidation [42,43]. Because lactate is also a major mitochondrial substrate in many tissues, and the metabolism of lactate and pyruvate may be interrelated, we reasoned that the metabolism of lactate could be also regulated by HKs and could involve, as proposed elsewhere, the malate aspartate shuttle (MAS) [25–27]. To test the hypothesis that lactate metabolism is regulated by HKs and involves the malate aspartate shuttle (MAS), we carried out experiments with the FRET-based sensor "Laconic" to monitor in real time the changes in cytosolic lactate in HEK cells in the absence of glucose to abolish the activity of the hexokinases. For reference, the FRET probe Laconic uses a lactate binding domain formed by the transcription factor

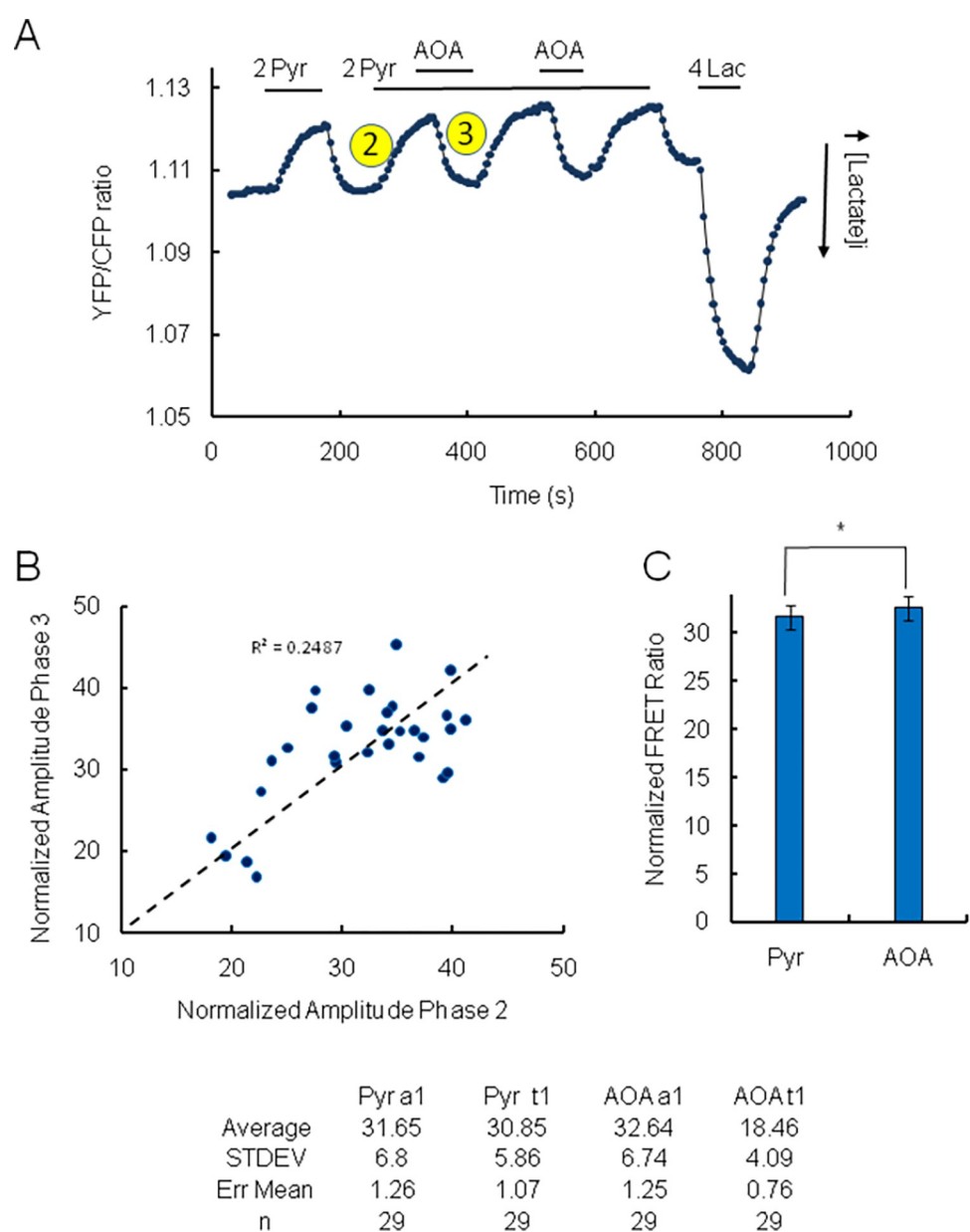

| | Pyr a1 | Pyr t1 | AOA a1 | AOA t1 |
|---|---|---|---|---|
| Average | 31.65 | 30.85 | 32.64 | 18.46 |
| STDEV | 6.8 | 5.86 | 6.74 | 4.09 |
| Err Mean | 1.26 | 1.07 | 1.25 | 0.76 |
| n | 29 | 29 | 29 | 29 |

**Fig 2. Addition of pyruvate in the absence of glucose decreases intracellular lactate level in wild-type HEK cells.** Inhibition of MAS by AOA reverses the effect of Pyruvate. Panel A. In this panel the ratio YFP/CFP is plotted as a function of time. An upward trend of the ratio corresponds to a decrease in intracellular lactate level and vice versa a downward trend indicates an increase in lactate level. At the end of the experiment, as in all experiments, 4 mM lactate was added to the perfusion solution. The resulting change in FRET ratio was used to normalize the FRET data. Panel A: Upon addition of 2mM pyruvate there is a rapid and sustained decrease in intracellular lactate level (phase 2). This decrease of lactate is fully reversible upon removal of pyruvate. Following addition of aminooxyacetate (AOA an inhibitor of malate aspartate shuttle via its aspartate aminotransferase component) in the presence of pyruvate, the decrease in in intracellular lactate is blocked. The effect of AOA is referred to as phase 3. The block was fully reversible upon removal of AOA. Panel B is a graph of the relationship between the amplitudes of phases 2 and 3, which shows a good correlation between the amplitude of the 2 processes. The dashed line represents a forced fit through the origin. Panel C shows a quantification of the reversal of the effect of pyruvate following addition of AOA. The left bar shows the normalized amplitude of the decrease in lactate evoked by the addition of pyruvate in the absence of glucose. In this case the normalized amplitude of the decrease in lactate level averaged 31.65±1.26 (n = 29). The right bar shows the normalized amplitude of the inhibition by AOA. In this case the mean amplitude was 32.64±1.25 (n = 29). A T test Two-Sample assuming equal variance returned a P value of 0.77 (*) suggesting no significant difference was observable between the two different interventions. The table shows the values used to generate the graph in panel C.

LldR bounded by the fluorophores mTFP (CFP) and Venus (YFP). As lactate binds to the sensor the energy transfer between the two fluorophores decreases and the ratio YFP/CFP decreases (Fig 1). In addition, this FRET sensor exhibits a nearly linear response between $10^{-5}$ and $10^{-2}$ M lactate [40]. Importantly our dose response curve, shown in Fig 1C, indicates that the changes in substrate that we record in our experiments fall within this range. The Kd derived from fitting our data to the intracellular *in vivo* lactate concentration is 1.26 mM. This compares to San Martin's in vitro value of 860 +/- 160 μM [40]. Based on these observations we estimate, for example, that NaCN-induced mitochondrial inhibition raises intracellular lactate up to 2 to 3mM.

We first investigated how addition of pyruvate affects intracellular lactate levels in the absence of glucose in wild-type HEK cells. Because lactate dehydrogenase (LDH) favors the transformation of pyruvate into lactate we expected that addition of pyruvate would elevate intracellular lactate level. However, we observed a robust decrease in intracellular lactate level following the addition of pyruvate. In this case the effect of 2 mM pyruvate was fast (τ = 30.85 ±1.07s; Fig 2) and sustained. The decrease in intracellular lactate level is denoted here as phase 2 (Fig 2A).

## The utilization of lactate by mitochondria depends on the activity of the malate aspartate shuttle (MAS)

To investigate the mechanism underlying the decrease in intracellular lactate level evoked by addition of pyruvate, we turned to some recent work. This postulates that generation of $NAD^+$ by the malate aspartate shuttle (MAS) in the mitochondrial intermembrane space, controls the activity of the lactate dehydrogenase (mLDH) to transform lactate into pyruvate in this compartment [25–27]. We hypothesize that a similar mechanism may account for our data. To test this hypothesis we used aminooxyacetic acid (AOA), an inhibitor of aspartate aminotransferase, which is an essential part of the MAS. Such inhibition prevents the reoxidation of cytosolic NADH by the mitochondrion [29,31,44]. Representative data shown in Fig 2A illustrates how addition of AOA blocks the decrease in intracellular lactate evoked by the addition of pyruvate (phase 3) within seconds (τ = 18.46±0.76s; Fig 2). Analysis of the data indicates that a correlation exists between the effect of AOA (increased lactate level, amplitude of phase 3) and the amplitude of phase 2 (Fig 2B).

Thus, addition of pyruvate to HEK cells with catalytically inactive hexokinases (no substrate present) evokes a decrease in the intracellular level of lactate, which is blocked by inhibition of MAS.

## The utilization of lactate by mitochondria depends on the activity of the malate aspartate shuttle (MAS) in the double HKI/HKII KD cells

Phosphorylation of glucose by hexokinases is the first step of glycolysis that leads to the generation of pyruvate. Thus, removal of glucose from the perfusion medium will prevent the formation of pyruvate. However, removal of glucose may have other effects on the cell metabolism that are not related to the role of HKs in glycolysis. To validate our data obtained with the removal of glucose in wild-type HEK cells, we repeated the same experiments in HEK cells in which both HKI and HKII expression had been markedly depleted using shRNAs (S6 Fig).

Representative data shown in Fig 3A1 and 3B1 corroborate the experimental results obtained with the addition of pyruvate in the absence of glucose. In the double HKI/HKII KD cells addition of pyruvate, now in the presence of 5mM glucose, caused a significant decrease in lactate characterized by an amplitude and a time course similar to those observed in wild-type HEK cells conducted in the absence of glucose. The normalized amplitude for the changes

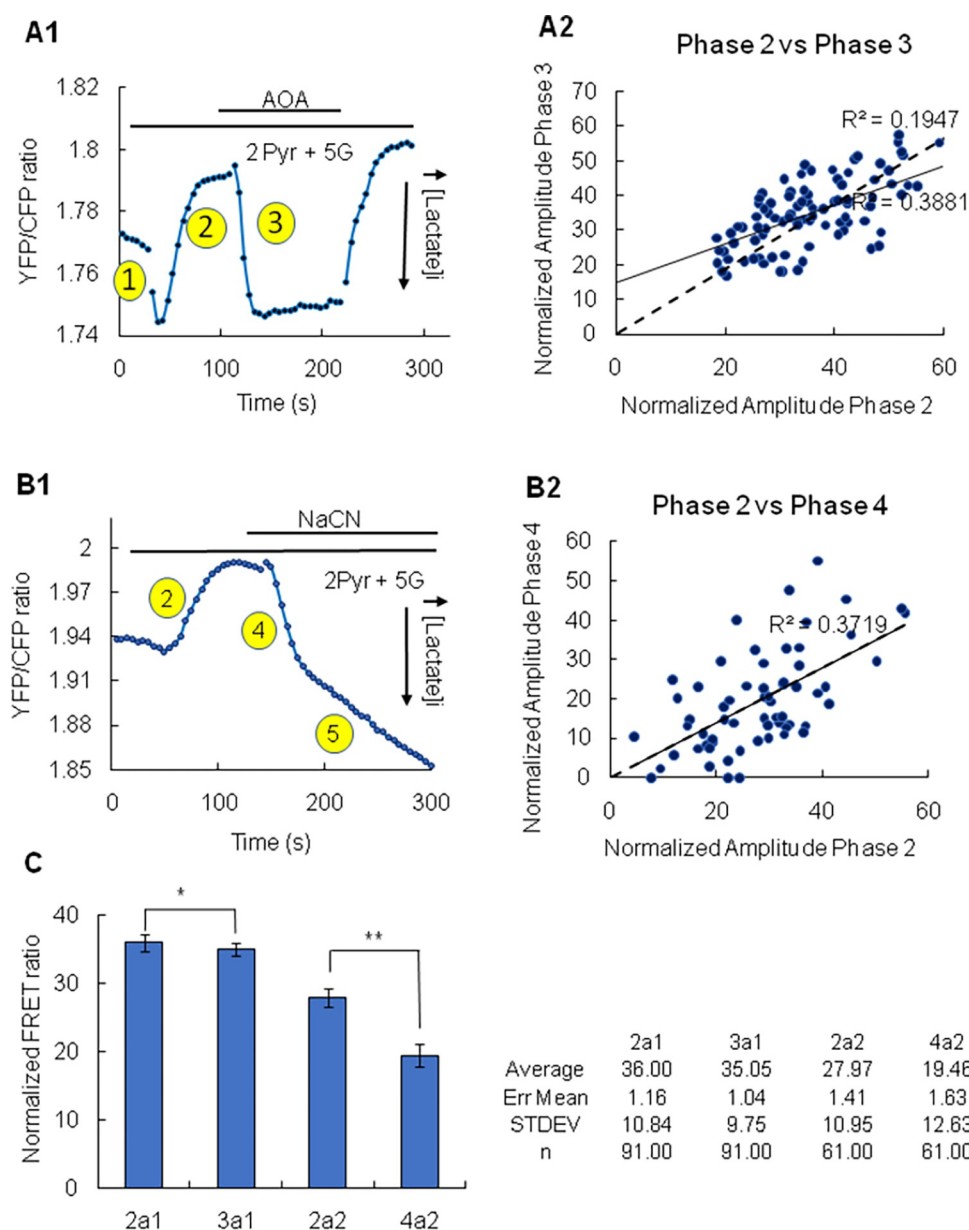

**Fig 3. The decrease in lactate induced by pyruvate is blocked by inhibition of MAS and of the TCA cycle in HKI/HKII double KD cells.** Panel A1 illustrates the inhibitory effect of aminooxyacetate (AOA), an inhibitor of the malate aspartate shuttle (MAS). In this case, addition of 2mM pyruvate caused first a brief increase in lactate (phase 1). This first phase was not always present, contrary to the sustained decrease in lactate (phase 2), which was always observed upon addition of pyruvate. Following the decrease in lactate evoked by pyruvate, addition of AOA reversed this effect, causing a rapid increase in lactate. This increase in lactate level is referred to as phase 3. The latter effect was fully reversible upon removal of AOA. Panel B2 is a graph of the relationship between the amplitudes of phases 2 and 3, which shows a good correlation between the amplitude of the 2 processes. The dashed line represents a forced fit through the origin. Panel B1 depicts a similar experiment but replacing aminooxyacetate (AOA) with NaCN. Again, upon addition of 2mM pyruvate there is a rapid and sustained decrease in intracellular lactate level (phase 2). This decrease is quickly reversed upon addition of NaCN an inhibitor of the electron transport chain (phase 4). This fast inhibitory phase was followed by a sustained one leading to intracellular lactate accumulation (phase 5). We will address the latter phase in a subsequent publication. Panel B2 is a graph of the relationship between the amplitudes of phases 2 and 4, which shows a good correlation between the amplitude of the two processes. Panel C shows a quantification of the reversal of the effect of pyruvate following addition of AOA and NaCN. The effects of AOA and NaCN were tested in two different sets of cells (1 and 2) at different times. The left part of the graph in Panel C is a quantification of the inhibitory effect of AOA. 2a1 is the normalized amplitude of the effect of pyruvate (phase 2) and 3a1 is the reversal of this effect following addition of AOA (phase 3). Phase 2a1 evoked by pyruvate had a mean

amplitude of 36.00±1.16 (n = 91). With the addition of AOA phase 3a1 had a mean amplitude of 35.05±1.04 (n = 91). The F test for variance homogeneity returned a value of 1.22. A T test Two-Sample assuming equal variance returned a P value of 0.25 (*). The right part of the graph in Panel C is a quantification of the effect of NaCN. 2a2 is the normalized amplitude of the effect of pyruvate (phase 2) and 4a2 is the reversal of this effect following addition of NaCN (phase 4). Phase 2a2 evoked by pyruvate had a mean amplitude of 27.97±1.41 (n = 61). With the addition of NaCN phase 4a2 had a mean amplitude of 19.6±1.63 (n = 61). The F test for variance homogeneity returned a value of 0.75. A T test Two-Sample assuming equal variance returned a P value <0.0005 (**). The table on the right of panel C shows the values used to generate the graphs on the left.

in lactate level was 36.00±1.16 with the double KD cells and 31.65±1.26 with the wild-type cells without glucose. It should be noted that in the double KD cells we observed in a few cells a transient increase in intracellular lactate level, lasting only a few seconds, upon addition of pyruvate (phase 1 Fig 3A1). Such an increase may reflect the transient transformation of pyruvate into lactate in the cytoplasm.of these cells. As with wild-type HEK cells, in the absence of glucose, addition of AOA in HKI and HKII KD cells blocks the decrease in intracellular lactate evoked by the addition of pyruvate (phase 3) within seconds (τ = 15.79±0.71s; Fig 5B). This value is similar to the rate measured with addition of AOA in wild-type HEK cells in the absence of glucose (τ = 18.46±0.76s; Fig 2). As for the wild-type cells, analysis of the data obtained with these double KD cells indicates that a correlation exists between the effect of AOA (increased lactate level, phase 3) and the amplitude of phase 2 (Fig 3A2). Furthermore, the plot in Fig 3C shows that there was no statistical difference between the amplitude of phase 2 and phase 3 in the double KD cells. These findings confirm our results obtained with wild-type HEK cells and the removal of glucose. Taken together these data support the hypothesis that cytoplasmic lactate is transformed into pyruvate by mLDH in the mitochondrial inter-membrane space via activation of the MAS and NAD$^+$ production (see Fig 4).

## The role of the TCA cycle in regulating the metabolism of pyruvate and lactate in the double HKI/HKII KD cells

To further investigate the role of mitochondrial metabolism in modulating the effect of pyruvate we investigated how inhibition of mitochondrial metabolism by the electron transport chain inhibitor NaCN affects the changes in lactate levels evoked by pyruvate. As shown in Fig 3B1 addition of NaCN blocked the utilization of lactate within seconds (τ = 20.54±1.45s; Fig 5B). We refer to this phase as phase 4. The plot in panel 3B2 indicate that a correlation exists between the amplitudes of the decrease in lactate induced by pyruvate (phase 2) and the increase induced by NaCN (phase 4), suggesting that the two phenomena may involve activation and inhibition of similar mechanisms linked to mitochondrial metabolism. However, the plot in Fig 3C shows that the amplitude of phase 2 and phase 3 were statistically different in this case. The smaller average amplitude of phase 4 compared to that of phase 2 may result from an underestimate of the amplitude of phase 4 as a double exponential must be used to fit simultaneously phases 4 and 5 (S1B Fig). This is because the fast increase in intracellular lactate evoked by the addition of NaCN (phase 4) is followed by a sustained increase in lactate in the continuous presence of NaCN (phase 5, see Fig 2A1). For clarity this presentation focuses on the fast phase of lactate accumulation evoked by NaCN (phase 4). The late phase (phase 5) is addressed in a separate publication.

The similarity between the inhibitory effects of NaCN and AOA suggests that they target related processes (see also Fig 9). This hypothesis is consistent with previous reports that the activity of the TCA cycle and MAS are interconnected [44,47,48].

Finally, these data obtained with the laconic FRET sensor in HEK cells have been validated using other cell types, such as HeLa cells (S2B Fig) and stem-cell-derived cardiomyocytes [49].

Results obtained with those cells indicate that NaCN-induced inhibition of the regulation of lactate metabolism by pyruvate is a process common to other cell types.

### Pyruvate carboxylation by the malic enzyme facilitates the activation of the TCA cycle and the utilization of lactate by mitochondria

Our data clearly show that lactate alone is not utilized by mitochondria unless pyruvate is added. Thus, additional pyruvate triggers or "primes" the transformation of lactate into pyruvate. As illustrated in the schematic in Fig 4 there are at least 3 pathways that may account for the effects of pyruvate. Pathway one, involves PDH and the transformation of pyruvate into acetyl CoA. Pathway two, involves the carboxylation of pyruvate into oxaloacetate (OAA). With a normal energy supply this transformation is modulated by pyruvate carboxylase (PC). However, as a result of energy deprivation pyruvate carboxylation pathway three may be

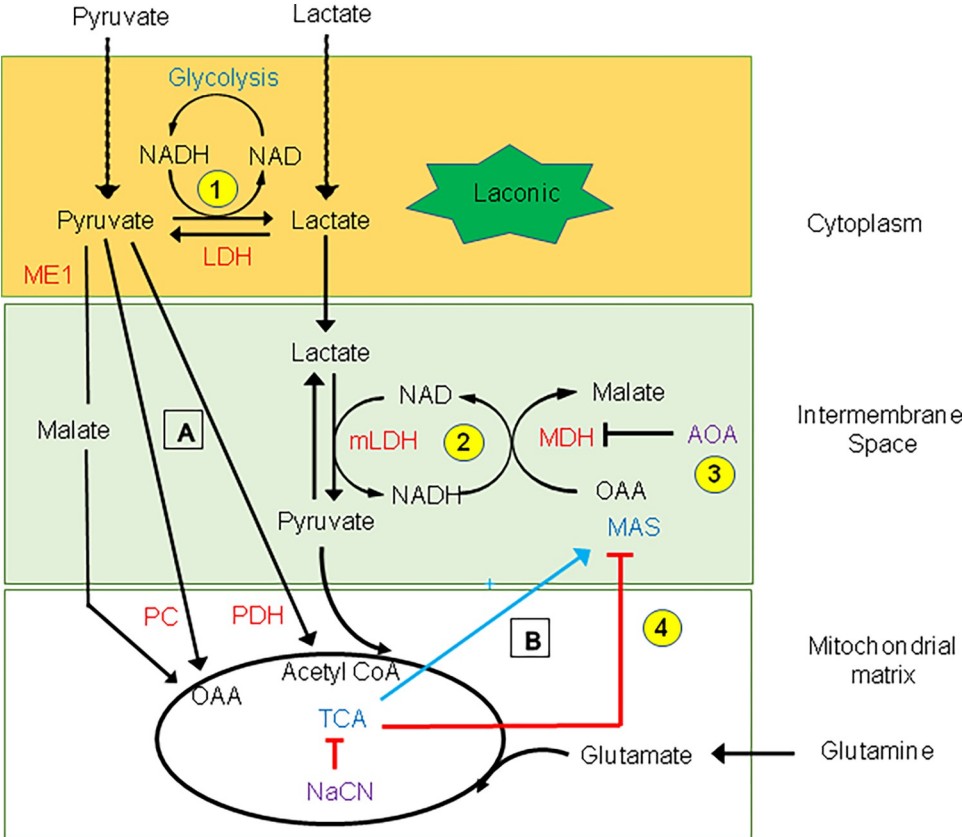

**Fig 4. Pyruvate-dependent regulation of lactate metabolism in HEK cells with HKs knocked-down.** Addition of cytosolic pyruvate activates the TCA cycle "A", which then activates the malate aspartate shuttle (MAS) "B" to facilitate the transformation of lactate into pyruvate in the mitochondrial intermembrane space "phase 2". This transformation of lactate into pyruvate is mediated via activation of mLDH (phase 2), resulting from the production of NAD⁺ by MAS. Pyruvate is then transported into the matrix to further supply the TCA cycle. Inhibition of MAS by AOA in the intermembrane space blocks the transformation of lactate into pyruvate "phase 3". Similarly, blocking of MAS as a result of the TCA cycle inhibition by NaCN stops the transformation of lactate into pyruvate in the intermembrane space "phase 4". In our schematic we show 3 different pathways that may account for the "priming" effect of cytosolic pyruvate on the TCA cycle "A". One involves pyruvate dehydrogenase (PDH) and the production of Acetyl CoA. The two others involve pyruvate carboxylation and the production of oxaloacetate (OAA). In one case OAA is directly generated via activity of the pyruvate carboxylase (PC), and in the other case malate is generated as an intermediary via the activity of the malic enzyme (ME1). Inhibition of the effect of pyruvate by an inhibitor of pyruvate carboxylation and of ME1 supports the latter pathway.

utilized via malic enzyme (ME1) to generate malate, which is then transported into the mito-
chondria to produce OAA [50]. To test these different pathways, we first used phenylpyruvate
an inhibitor of pyruvate carboxylation with HKI-HKII kd HEK cells. Data in Fig 5 shows that

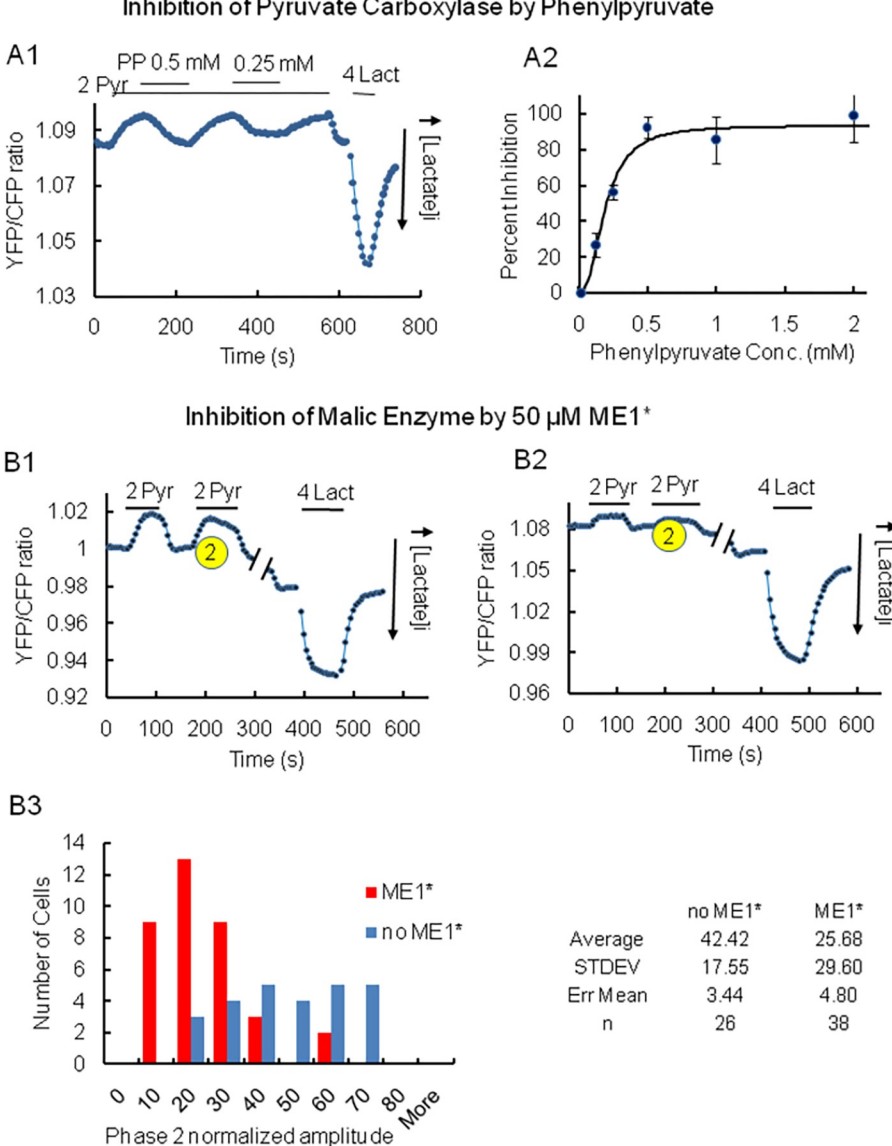

**Fig 5. A. Inhibition of the effect of pyruvate on lactate utilization by phenylpyruvate.** Panel A. Representative trace
of the inhibition of pyruvate-induced decrease in intracellular lactate by phenylpyruvate. Inhibition occurred with a
time constant of 63.7±5.5s (n = 13) upon addition of 0.5mM phenylpyruvate. The inhibitory effect of phenylpyruvate
was reversible at concentrations lower than 0.5mM, but poorly reversible at concentrations greater than 2mM. Panel B.
Increasing concentrations of phenylpyruvate were added in the presence of 2 mM pyruvate. The graph illustrates the
dose dependent inhibition by phenylpyruvate of lactate utilization (phase 2), which is expressed as percent inhibition.
Fitting of the data with a Hill equation yielded an IC50 of 0.24 mM. This value is comparable to the IC50 of 0.48 mM
obtained with purified pyruvate carboxylase [45,46]. B. Inhibition of the malic enzyme ME1 impairs pyruvate-induced
lactate utilization. Recordings in panel B are representative traces of the decrease in intracellular lactate level induced
by the addition of pyruvate in the double HKI-HKII kd cells under control conditions (B1) and after a 48hr incubation
with the inhibitor of ME1 (ME1*) at a concentration of 50μM (B2). The histogram in panel B3 illustrates the inhibitory
effect of ME1* on the amplitude of phase 2 (pyruvate-induced decrease in the level of lactate). The summary table
indicates that there is a statistical difference between the amplitude of phase 2 under control conditions and in the
presence of ME1*.

such an inhibition significantly decreases the amplitude of phase 2 (pyruvate-induced lactate utilization). This result suggests that the production of OAA due to pyruvate carboxylation, rather than oxidation due to PDH may be responsible for the effect of pyruvate. To further characterize the "priming" effect of pyruvate we used an inhibitor of the cytosolic malic enzyme (ME1). Incubation of the HKI-HKII kd HEK cells with 50μM ME1* (the inhibitor of ME1) for 48hrs prior to recordings significantly decreased the amplitude of phase 2 (Fig 5B). These data support the hypothesis that pyruvate carboxylation via ME1 plays a major role in pyruvate-induced lactate utilization by mitochondria.

## Modulation of pyruvate/lactate metabolism by hexokinases

Having established that the decrease in intracellular lactate level may be due to the MAS-dependent activity of mLDH and the transformation of lactate into pyruvate in the intermembrane space of mitochondria, we investigated how the expression of HKI or HKII could modulate this metabolism of lactate/pyruvate. We initially overexpressed either HKI or HKII, but since they had similar effects (S3 Fig lower panel), we focused on HKI, which is strongly bound to the OMM [35]. Focusing on HKI was also useful since the interaction of HKs with the OMM was a property that we intended to investigate.

Two types of change in lactate levels were observed in cells that overexpressed HKI and were cultured and subjected to similar induction protocols. The first showed the usual rapid decrease in intracellular lactate (phase 2), which occurs upon application of pyruvate to the bath, but with expression of HKI, this phase was now only transient, being followed by a delayed increase in lactate (compare Fig 6A and 6C). The amplitude of this delayed increase was variable, falling in some cases below base line (Fig 6C, phase 2d). In the second type of response, addition of pyruvate did not cause any initial decrease in lactate (no phase 2), but there was instead an instantaneous and sustained increase in lactate, corresponding to a phase 1 (Fig 6D).

Furthermore, in the presence of HKI the effect of NaCN was also transient or non-existent. Indeed, when addition of pyruvate evoked a transient decrease in lactate (phase 2d), the subsequent addition of NaCN evoked an increase in lactate, but this phase was also transient (Fig 6C, phase 4d). However, when addition of pyruvate did not evoke a decrease in lactate (no phase 2, but phase 1), the subsequent addition of NaCN had no further effect (Fig 6D). These data suggest that expression of HKs blocks the activity of the MAS to transform lactate into pyruvate, and this effect is either instantaneous or delayed.

Thus, on average overexpression of HKI decreases the amplitude of phase 2 from 27.96±1.4 to 8.55±1.37 (Fig 7A). These data demonstrate that HKI's activity is a potent inhibitor of the regulation of lactate metabolism by pyruvate.

## Modulation of pyruvate/lactate metabolism by hexokinases: The role of PPP and NAD(P)H

It has been acknowledged for many years that the regulation of mitochondrial metabolism by HKs is linked to their catalytic activity [36,51]. Based on this observation, it is inferred that regulation by HKs may involve the utilization of ATP, subsequent production of ADP, or the production of G6P, the latter controlling not only glycolysis but also the pentose phosphate pathway (PPP). Activation of PPP is one of the main sources of NAD(P)H and may, upon activation, interfere with the regulation of the NADH-dependent mLDH in the intermembrane space as illustrated in Fig 8. To test this hypothesis, we investigated how 6-aminonicotinamide (6-AN), an inhibitor of glucose 6 phosphate dehydrogenase (G6PD) [52,53] modulates the inhibitory effect of HKs. The effect of 6-AN on the production of lactate in HEK cells

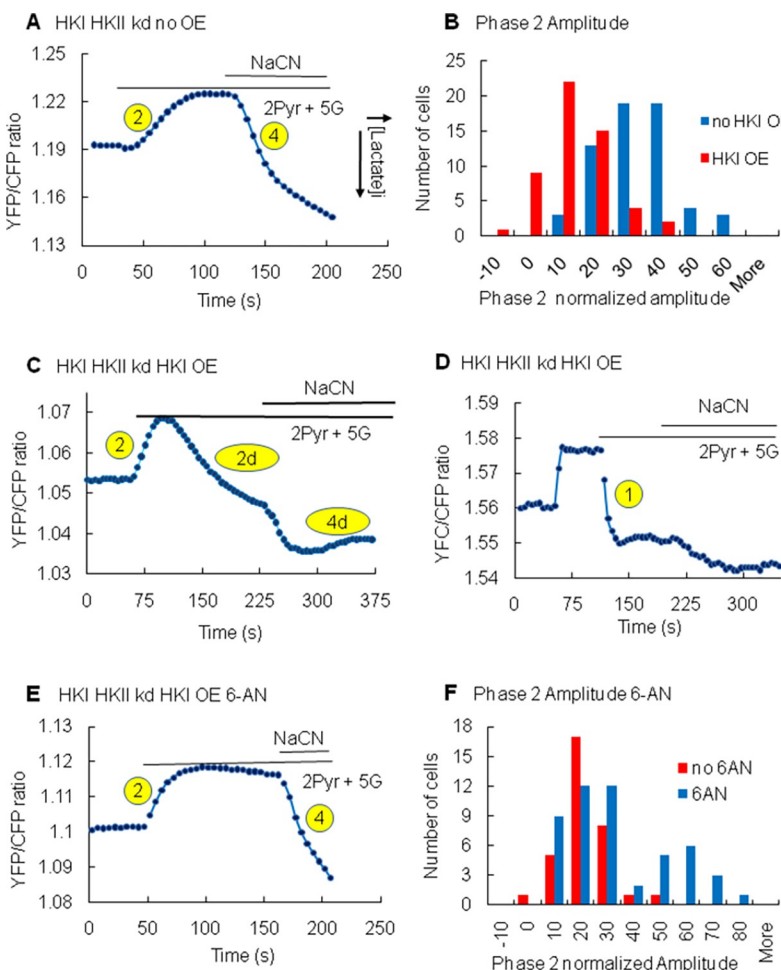

**Fig 6. Effects of HKI overexpression and 6-AN addition on pyruvate-induced increase in lactate metabolism.** As a reference, panel A illustrates the decrease in intracellular lactate evoked by addition of pyruvate (phase 2) and inhibition by NaCN (phase 4). As a reminder, increase in lactate level is shown as a downward deflection in this panel as well as the other panels. Panels C and D illustrate 2 types of inhibition of pyruvate-mediated lactate metabolism due to overexpression (OE) of HKI. In one case (panel C) the inhibition was delayed and led to a clear phase 2 followed by a gradual increase in lactate (phase 2d). In the other case (panel D) inhibition was present right at the beginning and there was no phase 2. The lack of phase 2 in this case uncovered a sustained increase in lactate, corresponding to a sustained phase 1. The graphs in Panel B represent a quantification of the inhibitory effect of HKI on the amplitude of phase 2 with a mean amplitude for phase 2 of 27.96±1.4 (n = 61) in the absence of HKI overexpression and 8.55±1.37 (n = 53) with HKI overexpression. The trace in Panel E illustrates how incubation with 6-AN, the inhibitor of glucose-6-phosphate dehydrogenase (G6PD), completely reverses the inhibition due to HKI overexpression with clear phases 2 and 4 now present. The graphs in panel F illustrate this reversal of inhibition evoked by 6-AN. The mean value for phase 2 amplitude in the absence of 6-AN, but with HKI overexpressed, was 15.64±2.72 (n = 33) and 30.0±4.16 (n = 52) in the presence of 6-AN with HKI overexpressed. It is clear from this graph that while phase 2 was reactivated by 6-AN in some cells, it was not in others.

expressing HKI was often robust and dependent on the time of incubation, with a maximum effect observed with time of incubation greater than 24hrs. Indeed 6-AN prevented the inhibitory effect of HKs overexpression, with addition of pyruvate causing a significant and sustained decrease in intracellular lactate level in many cells (Fig 6E). With addition of 6-AN in the presence of HKI the amplitude of phase 2 increased on average from 15.64±2.72 to 30 ±4.16 (Fig 5A).

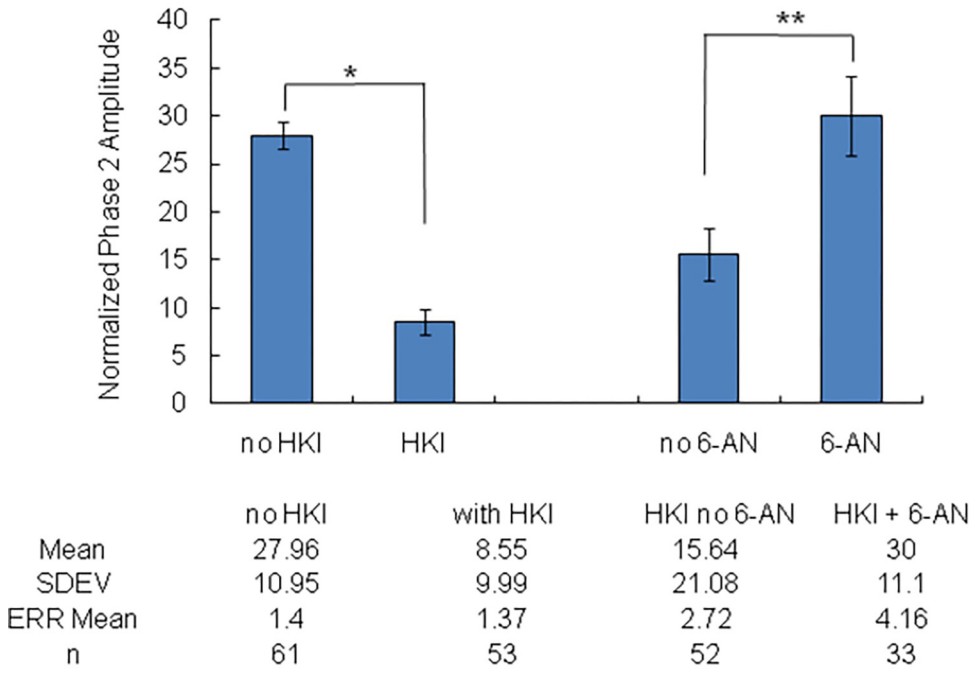

| | no HKI | with HKI | HKI no 6-AN | HKI + 6-AN |
|---|---|---|---|---|
| Mean | 27.96 | 8.55 | 15.64 | 30 |
| SDEV | 10.95 | 9.99 | 21.08 | 11.1 |
| ERR Mean | 1.4 | 1.37 | 2.72 | 4.16 |
| n | 61 | 53 | 52 | 33 |

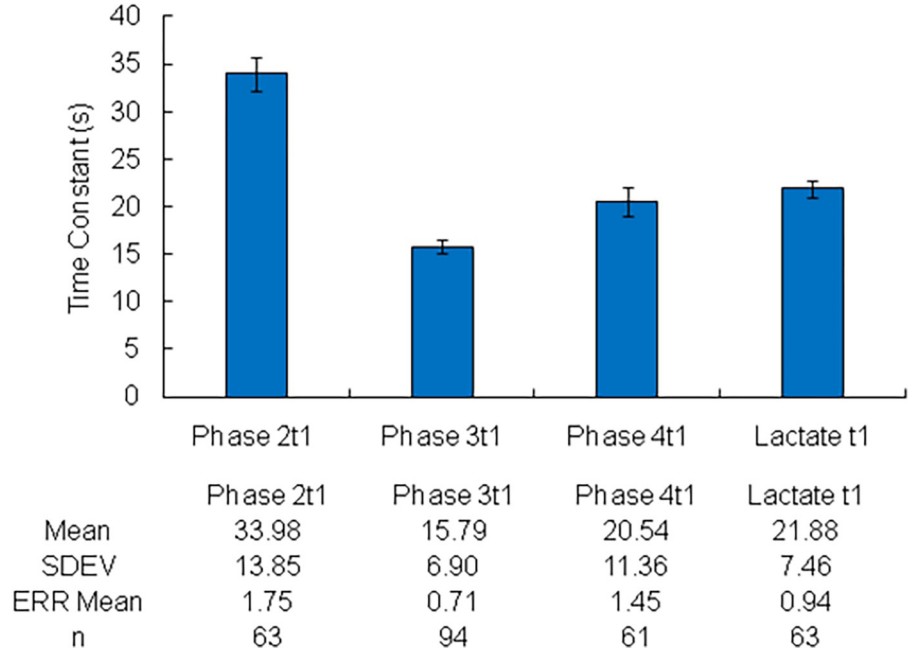

| | Phase 2t1 | Phase 3t1 | Phase 4t1 | Lactate t1 |
|---|---|---|---|---|
| Mean | 33.98 | 15.79 | 20.54 | 21.88 |
| SDEV | 13.85 | 6.90 | 11.36 | 7.46 |
| ERR Mean | 1.75 | 0.71 | 1.45 | 0.94 |
| n | 63 | 94 | 61 | 63 |

**Fig 7. A. Inhibition of phase 2 amplitude by HKI OE and its reversal by 6-AN.** The left part of the graph in Panel A represents a quantification of the inhibitory effect of HKI on the amplitude of phase 2 with a mean amplitude for phase 2 of 27.96±1.4 (n = 61) in the absence of HKI overexpression and 8.55±1.37 (n = 53) with HKI overexpression. A Shapiro-Wilk test returned a P value of 0.14 (no HKI OE) and P = 0.2 (HKI OE). The F test for variance homogeneity returned a value of 0.25. A T test yielded a P value <0.0005 (*). The right part of the graph in Panel A illustrates the

reversal of inhibition evoked by 6-AN. The mean value for phase 2 amplitude in this case, the absence of 6-AN, but with HKI overexpressed, was $15.64\pm2.72$ (n = 33) and $30.0\pm4.16$ (n = 52) in the presence of 6-AN with HKI overexpressed. The F test yielded $p<0.0005$ and the t-test (unequal variance) returned $P <0.005$ (**). The Shapiro-Wilk test returned P of 0.01 for HKI OE no 6-AN and P of 0.07 for HKI OE plus 6-AN. B. Time constant for the changes in FRET ratio evoked by addition of pyruvate, AOA and NaCN. Phase 2t1 is for the addition of pyruvate, phase 3t1 for the addition of AOA, Phase 4t1 for the addition of NaCN and lactate t1 for the addition of lactate.

Based on these observations, we hypothesize that HKs block pyruvate-induced utilization of lactate by blocking the transformation of lactate into pyruvate by mLDH in the intermembrane space as a result of NAD(P)H elevation (see schematic in Fig 8).

## Discussion

Monitoring FRET ratios with a probe that is specific for lactate has allowed us to move the measurement of intracellular metabolites from a flux or stop flow analysis to real-time measurements. The fact that many of the changes in intracellular FRET ratios occurred within seconds and were often transient makes this FRET sensor a unique tool to study changes in cytoplasmic metabolites.

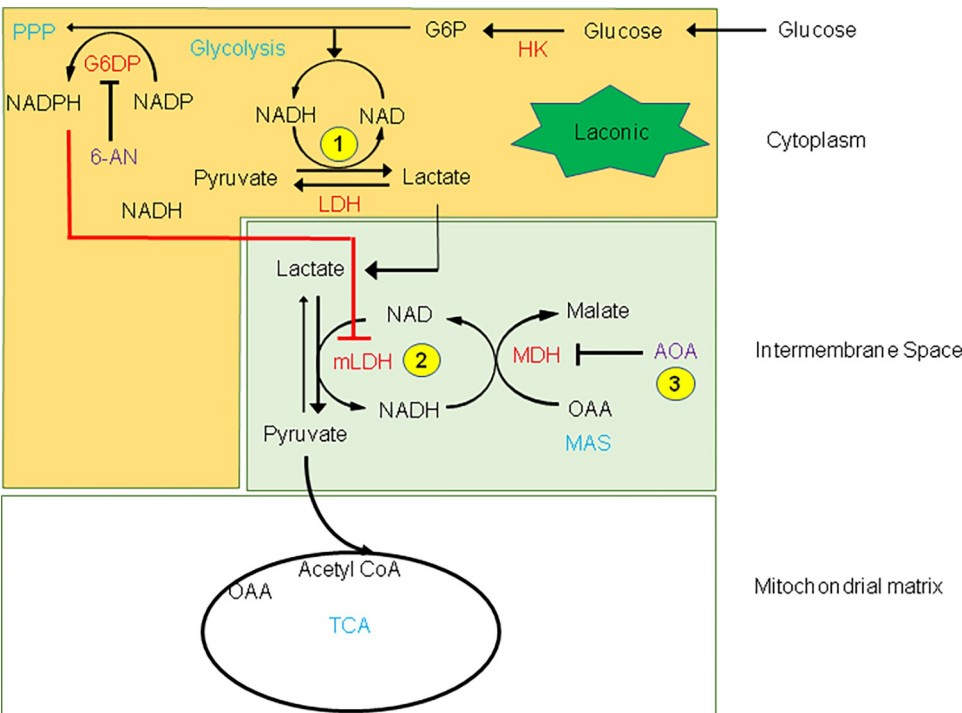

**Fig 8. Effects of HK overexpression on pyruvate-dependent lactate metabolism in HEK cells.** As illustrated in Fig 3 the malate aspartate shuttle (MAS) may facilitate the transformation of lactate into pyruvate, which is then transported across the inner mitochondrial membrane. The present schematic illustrates how HKs may inhibit this process. Central to this process is the role of mLDH in the intermembrane space and its regulation by $NAD^+$ to transform lactate into pyruvate during phase 2. We hypothesized that HKs block the transformation of lactate into pyruvate by inhibiting mLDH, a process that would involve the production of NADPH by G6PD of the pentose phosphate pathway (PPP). This hypothesis is supported by our data obtained with 6-AN, the inhibitor of G6PD, in HKI expressing HEK cells. In this case, incubation with 6-AN prevents the production of NADPH by G6PD, thereby relieves the inhibition of mLDH. The latter results in a facilitated transformation of lactate into pyruvate. In some instances, the inhibition of lactate metabolism by HKI was delayed (phase 2d) reflecting perhaps the transformation of NADPH into NADH by the nicotinamide nucleotide transhydrogenase (NNT) that converts $NADPH/NAD^+$ to $NADP^+/NADH$ [54].

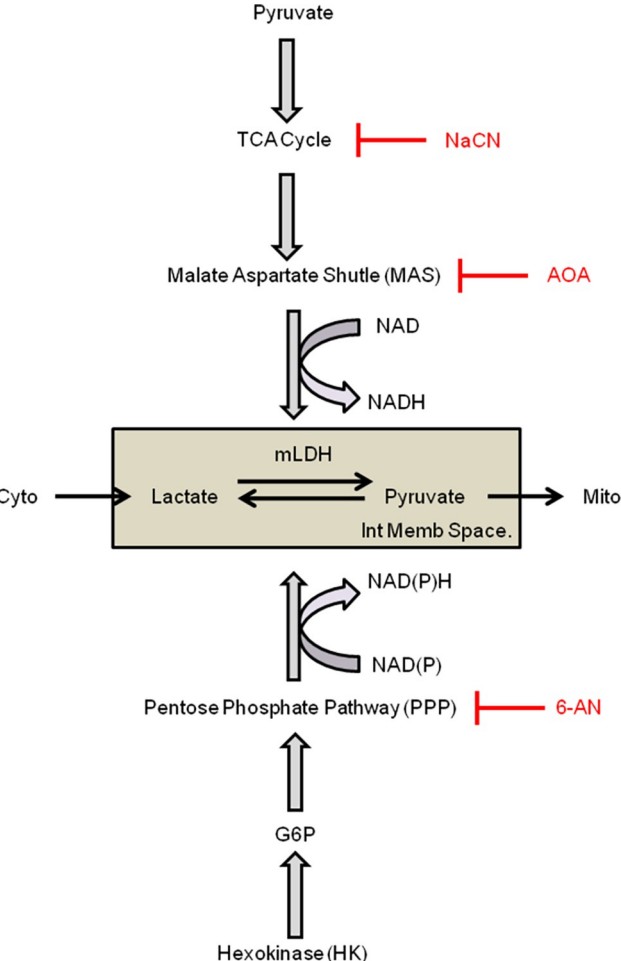

**Fig 9. Pathways influencing the forward and reverse Lactate to Pyruvate reaction.** Driving the forward reaction: Pyruvate activates the TCA cycle, which activates the malate aspartate shuttle (MAS). The production of NAD by MAS activates mitochondrial-associated lactate dehydrogenase (mLDH) in the intermembrane space of the mitochondria (grey box) to facilitate the transformation of lactate into pyruvate. This process is inhibited by NaCN, which blocks the TCA cycle and by aminooxyacetate (AOA), which blocks MAS. Driving the reverse reaction: Hexokinase (HK) blocks the transformation of lactate into pyruvate via production of glucose 6 phosphate (G6P), which facilitates the generation of NAD(P)H by the pentose phosphate pathway. NAD(P)H reverses the lactate to pyruvate reaction.

The main observations that we have made are summarized in the Fig 9 diagram and are as follows: 1) In normal cells upon removal of hexokinase substrate- glucose- and in genetically modified cells without HKs, the addition of pyruvate causes a decrease in cytoplasmic lactate levels within seconds. 2) AOA an inhibitor of the Malate Aspartate Shuttle (MAS) blocks the decrease in cytoplasmic lactate. 3) The effect of pyruvate is also blocked by the mitochondrial inhibitor NaCN, which acts to block the TCA cycle. 4) Pyruvate-induced lactate diminution is blocked by overexpression of HKs and the latter effect is inhibited by the pentose phosphate pathway (PPP) inhibitor 6-AN.

## The transformation of lactate into pyruvate requires pyruvate-dependent activation of the TCA cycle and the subsequent activation of MAS

Addition of pyruvate to HEK cells in which HKs were knocked down (HK KD) resulted in a dramatic decrease in intracellular lactate level. This was unexpected since the equilibrium

between pyruvate and lactate in the cytosol is shifted towards lactate. What was also surprising was the complete block of this effect by aminooxyacetic acid (AOA). Both pyruvate-induced decrease in lactate and its reversal by AOA occurred within seconds and were fully reversible upon removal of the metabolite, or of the inhibitor. AOA inhibits the aspartate aminotransferase, a PLP-dependent enzyme, which is an essential part of the malate-aspartate shuttle (MAS), and is located in part in the mitochondrial intermembrane space [2,29–31]. An important role of MAS is to facilitate the reoxidation of cytosolic NADH produced by the mitochondria. It has been suggested that the resulting increase in $NAD^+$ allows for the transformation of lactate into pyruvate in the intermembrane space via increased activity of the lactate dehydrogenase present in this compartment (mLDH) [25,26]. Accordingly, inhibition of MAS by AOA would cause a decrease in $NAD^+$ in the mitochondrial intermembrane space, blocking the transformation of lactate into pyruvate by mLDH.

Once lactate is transformed into pyruvate, the latter is moved into the matrix by the specific mitochondrial carriers MPC1 and MPC2 [19,20]. It has been suggested that for passive entry of pyruvate into the matrix to occur, activation of the TCA cycle is necessary to allow for the transformation of pyruvate into acetyl CoA, which creates a gradient for pyruvate [25]. The transformation of lactate into pyruvate is necessary, as the mitochondrial inner membrane is impermeable to lactate. Therefore, mLDH's role is to help replenish or maintain the pyruvate gradient to diffuse into the matrix. While we certainly agree with this concept our data also strongly support the involvement of a limiting step that regulates the utilization of lactate by mitochondria. This step involves activation of the MAS by the TCA cycle as illustrated in Fig 4. Such a coupling between the activity of MAS and that of the TCA cycle has been previously reported [44,47,48]. Thus, lactate alone cannot be metabolized and taken up by mitochondria unless pyruvate is present to diffuse into the matrix, generate acetyl-CoA via pyruvate dehydrogenase (PDH), or oxaloacetate via carboxylation of pyruvate, and activate the TCA cycle. Only then can activation of MAS take place to facilitate the transformation of lactate into pyruvate. Our data obtained with the pyruvate carboxylase inhibitor, phenylpyruvate, [45,46] (Fig 5A) indicates that "priming" by pyruvate involves pyruvate carboxylation rather than pyruvate dehydrogenase (PDH). In addition, pyruvate carboxylation may result from either the activity of pyruvate carboxylase (PC) or of the malic enzyme (ME1) when the cell's energy level drops [50]. To better differentiate between these two pathways, we used a specific inhibitor of ME1 (ME1*). Incubation with this inhibitor for 48 hrs decreased the amplitude of phase 2, suggesting impairment of pyruvate-induced lactate utilization. This result favors the hypothesis that transformation of pyruvate into malate and then OAA is part of the mechanism that primes the TCA cycle. To summarize: the production of oxaloacetate replenishes the TCA cycle's substrates, signaling activation of the TCA cycle, increasing the utilization of acetyl-CoA and therefore the activity of PDH to facilitate the transformation of lactate into pyruvate (see Fig 4). In this case, the utilization of acetyl-CoA is referred to having a "sink effect".

Our finding that addition of the mitochondrial inhibitor NaCN blocked pyruvate-induced decrease in lactate level within seconds, supports the hypothesis whereby both activation of MAS and of the TCA cycle may be necessary for transformation of lactate into pyruvate. In our model NaCN blocks pyruvate utilization by inhibiting mitochondrial function and this step precedes lactate transformation into pyruvate (Fig 4). Consideration of this sequence of events leading to the regulation of lactate metabolism will become relevant when we discuss later how this phenomenon relates to known regulatory processes.

### HKs block pyruvate-induced decrease in cytosolic lactate levels in an NADH-dependent manner

Our data show that overexpression of HKI or HKII blocks pyruvate-induced decrease in cytosolic lactate level in either an instantaneous or a delayed manner. We have proposed that pyruvate-induced decrease in cytosolic lactate occurs upon activation of MAS which activates mLDH in the intermembrane space in response to $NAD^+$ production. To account for this process, it could be postulated that inhibition of pyruvate-induced decrease in cytosolic lactate by catalytically active HKs involves NADH produced during glycolysis by GAPDH. Such increase in NADH would inhibit mLDH in the mitochondrial intermembrane and prevent the transformation of lactate into pyruvate. As illustrated in Fig 6, our data obtained with 6-aminonicotinamide (6-AN) an inhibitor of the pentose phosphate pathway (PPP) suggests otherwise. The role of catalytically active HKs is to phosphorylate glucose into G6P, which is the branching point for the glycolysis catabolic pathway and the PPP anabolic pathway. As a result the activity of HKs may indeed facilitate the production of NADH by glycolysis, but would also facilitate the transformation of NADP into NADPH during the first step of the PPP as glucose 6 phosphate dehydrogenase (G6PD) oxidizes G6P [55]. Note that the PPP is one of the main sources of NADPH production in cells [56]. Our data shows that 6-AN, a specific inhibitor of G6PD, efficiently blocks the inhibitory effect of catalytically active HK, by preventing pyruvate-induced decrease in lactate. This observation suggests that NADPH production by the PPP accounts for most of the effect of HK on pyruvate-induced decrease in lactate level. For this process to occur we must assume that an increase in NADPH is associated with an elevation of NADH, which is the coenzyme regulating the activity of MAS. One possible mechanism accounting for the transformation of NADPH into NADH may involve the nicotinamide nucleotide transhydrogenase (NNT) that converts $NADPH/NAD^+$ to $NADP^+/NADH$ by catalyzing hydride transfer from NADPH to $NAD^+$ [54] at the inner mitochondrial membrane [57]. Activation of any of these steps could account for the delayed responses observed during phase 2d and 4d.

### Pathophysiological relevance of MAS inhibition

Our data obtained with AOA and the regulation of MAS corroborate previous studies of cardioprotection, using ischemic preconditioning (IP) maneuvers in heart. Ischemic preconditioning is a complex protective process that involves treating, for instance, cardiac muscle with multiple short episodes of an injuring maneuver prior to the occurrence of the main injury. Ischemia is one such example where short ischemic episodes lead to protection against ischemic injury. Accordingly, short application of the MAS inhibitor AOA has been shown to be cardioprotective [31,58], even though AOA itself causes a reduction of cytosolic $NAD^+$, lowers energy levels in the perfused working heart, which results in failure of the myocardium [30,31]. Thus, short applications of AOA would be protective as they transiently inhibit MAS and mitochondrial reoxidation of cytosolic NADH. Our data suggest that inhibition of pyruvate-induced lactate utilization by AOA as a result of $NAD^+$ lowering may play a role in AOA-induced cardioprotection.

### The dual regulation of pyruvate and lactate metabolism by HKs and its Physiological relevance

Both pyruvate and lactate may be beneficial as mitochondrial substrates during glycolysis downregulation [59,60]. However, our data suggest that even though the metabolism of these two substrates may be interconnected, they are likely to differ. Indeed, according to our

scheme, and as previously shown by others, pyruvate utilization is directly regulated by pyruvate dehydrogenase (PDH) to generate acetyl CoA and activate the TCA cycle. HKs regulate this process via production of ADP, which then blocks pyruvate–dehydrogenase- kinase (PDK) and activates PDH [61–63]. On the other hand, the regulation of lactate metabolism by HKs would require the activation of the PPP, which would elevate intracellular NADPH, and in turn inhibit MAS to prevent the transformation of lactate into pyruvate by mLDH. Thus, the regulation of lactate metabolism by HKs requires the activation of MAS in addition to the transformation of pyruvate into acetyl CoA. This additional step involving MAS may account for the differential regulation of pyruvate and lactate metabolisms by HKs, and by insulin for that matter, since regulation by insulin may involve the interaction of HKs with mitochondria [11,64].

Such differential regulation is not uncommon and has already been reported for the regulation of glucose and fatty acids oxidation by mitochondria. Indeed, HKs interaction with OMM stimulates in an insulin-dependent manner the use of pyruvate by mitochondria, while blocking the oxidation of fatty acids. Even though these two substrates enter the TCA in a similar location they do so via two pathways that are differently regulated by insulin and/or interaction of HKs with OMM. This dual process is the basis for the switch in energy substrate observed in the normal heart with insulin and in the failing heart [6,7,10,65]. Because lactate utilization is also downregulated by HKs, we postulate that this process is part of the switch in substrate modulated by HKs. In other words when pyruvate metabolism is upregulated by insulin and HKs, both fatty acids and lactate metabolisms may be downregulated.

## Conclusions and future directions

Our studies bring to light some new mechanisms to explain how mitochondria regulate the metabolism of pyruvate and lactate and how interaction of HKs with OMM affects these processes. Specifically, they suggest that lactate is used by mitochondria and that this utilization requires the activation of the TCA cycle by pyruvate. This process is facilitated by the activity of MAS which generates $NAD^+$ and activates mLDH in the mitochondrial intermembrane space to transform lactate into pyruvate.

Interaction of HKs with mitochondria also blocks pyruvate-induced lactate utilization by mitochondria as a result of MAS inhibition. On this basis we propose that the switch in energy metabolism modulated by interaction of HK with mitochondria, which involves increased glucose oxidation and decrease fatty acid oxidation, is also accompanied by a decrease in lactate utilization by mitochondria.

Our studies suggest a number of directions for future experiments. For instance, we propose that HKs are likely to regulate the activity of MAS via production of NAD(P)H. It follows, assuming that HKs must interact with mitochondria for this process to take place that NAD(P)H may accumulate in a specific compartment near MAS. As of yet, the techniques to image NAD/NADH are not sensitive enough to make measurements in such a small compartment, the intermembrane space, to directly answer this question. However, we have already used fluorescent probes with single mitochondria in vivo, and with the perfecting of the FRET probes these types of measurements may soon become feasible.

Also, we have used in the present study some pharmacological agents to test a number of molecular pathways that may be involved in the regulation of lactate and pyruvate metabolism by HKs. We acknowledge that our conclusions would benefit from further experiments, since pharmacological agents are not always as specific as we would like them to be. It happens that the pathways that we tested involve a number of enzymes, G6PD, ME1 and GOT1 (Glutamic-oxaloacetic transaminase), that have already been knocked down in HEK cells. Using these cell lines, once they are more available, could greatly strengthen our hypotheses.

## Supporting information

**S1 Fig. Lactate dose response curve and Curve fitting.** Panels A1 and A2 illustrate a dose response using the FRET sensor Laconic expressed in HEK cells. The cells were exposed to increasing concentrations of lactate as shown in panels A1. The change in FRET ratio was then plotted as a function of concentration and fitted with a Hill equation. The fits yield a Kd of 1.26 mM. This value obtained in intracellular milieu is close to that reported by San Martin et al. in 2013 for in vitro experiments (830±160 μM). Because the FRET ratio values reported in our results are expressed as a percentage of the maximal change obtained at saturating concentration of substrate, these values may be used to get an estimated of the changes in concentration observed under the various experimental conditions. Panels B, C, and D are curve fittings of changes in FRET ratio. In this figure and all the other figures a downward trend in FRET ratio indicates an increase in intracellular lactate. Panel B the increase in intracellular lactate levels evoked by NaCN "phases 4 and 5" were fitted with a sum of two exponential functions. The amplitudes and time constants of the two phases were derived from the fit. In C and D the changes evoked by NaCN with HK overexpression were best fitted by a combination of an exponential and sigmoidal functions. In this case the effects of NaCN on the amplitude of phases 2 and 4 were estimated as the difference between the beginning and end of the fitted traces. When the value at the end of the trace exceeded that at the beginning a negative value for the amplitude was derived from the calculation.
(TIF)

**S2 Fig. A Fig Experimental variability in wild-type HEK293T cells.** Changes in intracellular concentration of lactate measured using the FRET sensor laconic expressed in wild-type HEK cells. A decrease in FRET ratio reflects an increase in intracellular lactate. These 2 panels illustrate very different response to 2 mM NaCN in 2 adjacent cells. In the left panel addition of pyruvate caused a decrease in lactate and the addition of 2 mM NaCN resulted in a pronounced increase in intracellular lactate level. In contrast the trace in the right panel show that addition of pyruvate caused a transient decrease in lactate and addition of NaCN had little effect in this condition. Panel B. Changes in intracellular concentration of lactate measured using the FRET sensor laconic expressed in wild-type HeLa cells. In HeLa cells as in HEK cells (upper left panel) addition of pyruvate caused a decrease in lactate and the addition of 2 mM NaCN resulted in a pronounced increase in intracellular lactate level. This data validates our measurements made in HEK cells.
(TIF)

**S3 Fig. Upper panel.** Knocking down only HKII or HKII plus HKI has similar effect on lactate accumulation as a result of mitochondrial inhibition by NaCN. In these experiments there was no further expression (no OE) of HKI or HKII following DOX-induced knock down. For the HKI HKII KD (no OE) the mean normalized amplitude for phase 5 was 64.4 ± 3.55 (n = 63). For HKII KD (no OE) the mean phase 5 normalized amplitude was 71.6 ± 5.08 (n = 21). F-test yielded a P value of 0.12. The t-test (equal variance) P value was 0.97. The shapiro-wilk test P value for HKI HKII KD no OE was 0.1, and 0.07 for HKII. These results support the hypothesis that HKII is the main hexokinase expressed in wild type HEK cells. Lower panel: Overexpression of HKI or HKII has similar effects on lactate accumulation as a result of mitochondrial inhibition by NaCN. These histograms depict the effects of the overexpression (OE) of HKI and of HKII in HEK cells in which HKI and HKII had been previously knocked down. For HKI OE in HEK cells in which HKI HKII had been previously knocked down (KD) the mean normalized amplitude for phase 5 was 16±1.8 (n = 55). For HKII OE in HKI HKII KD HEK cells the mean normalized amplitude for phase 5 was 23.4±3.8 (n = 27). An F-test yielded a P

value of 0.012 and a t-test (unequal variance) yielded P<0.0005. The shapiro-wilk test P value for HKI OE was 0.07, and 0.9 for HKII OE. These results strongly suggest that overexpression of HKI or HKII has similar effects on the regulation of lactate metabolism by mitochondria. (TIF)

**S4 Fig. Upper graph.** Box-and-whisker plots for the amplitude of phase 2, 3 and 4. It should noted that the 2 plots on the left (2a1, 3a1) and the 2 plots on the right (2a2, 4a2) show the results of two separate sets of experiments in which AOA (2a1, 3a1) and NaCN (2a2, 4a2) were tested. The values shown in the table below the graph were derived from the data used to generate the above plots. Lower graph, box-and-whisker plots for the amplitude of phase 2 without and with overexpression of HKI, and with and without 6-AN in the presence of HKI overexpression. The values shown in the table below the graph were derived from the data used to generate the above plots. The horizontal lines mark the median, the box limits indicate the 25th and 75th percentiles. Outlier points are included. It is to be noted that we report some variability in responses of HEK cells. These cells are clonal and this analysis indicates that even using such "identical" cells variance can be seen. (TIF)

**S5 Fig. Box-and-whisker plots for the time constants of phases 2, 3, 4 and addition of lactate.** the horizontal lines mark the median, the box limits indicate the 25th and 75th percentiles. The values shown in the table below the graph were derived from the data used to generate the above plots. (TIF)

**S6 Fig. Upper and middle panels.** Uncropped gels representing HKI, HKII and HKI/HKII knocked down in HEK cells. Columns 3, 5 and 8 correspond to control conditions without doxycycline. Column 2 is for HKI KD, column 6 for HKII KD and column 8 for the double HKI/HKII KD. Refer also to Fig 1 Panel C for lane labeling. Lower panel: Gel analysis for the 3 conditions depicted above (HKI KD, HKII KD and HKI/HKII KD). The graph was generated using the lane profile function of Image J. Control values (no doxycycline) are shown to the right for each condition. (TIF)

**S7 Fig. Data generated with studies of Aminooxyacetic acid (AOA).** In this Excel sheet, as with the following Excel sheets, the first column corresponds to the cell number. In columns "Phase 2a, 3a, 4a and 5a" the values correspond to the amplitude of phase 2, 3, 4 and 5 derived from the fit of the FRET traces. In columns "Phase 2t1, 3t1, 4t1 and 5t1" the values correspond to the time constant of Phase 2, 3, 4 and 5 derived from the fit of the FRET traces. Values in the columns labeled "lactate a" and "lactate t1" correspond to the amplitude and time constant of the FRET signal generated by the addition of 4mM lactate. Values in columns labeled "Norm 2a, 3a, 4a and 5a" were generated by normalizing the values for 2a, 3a, 4a and 5a to their corresponding value obtained upon addition of lactate in the same cell. Each row in this spread sheet correspond to one cell. Whole numbers for the time constants result from a fit of the data that exceeded the limits provided during the fit. (XLSX)

**S8 Fig. Data obtained in experiments carried out in HEK cells in which HKI and HKII were knocked down and with no overexpression of either HKI or HKII.** This worksheet shows a graph of the amplitude of phase 4 vs that of phase 2. (XLSX)

**S9 Fig. Data obtained in experiments carried out in HEK cells in which HKI and HKII were knocked down and in which HKI was overexpressed.**
(XLSX)

**S10 Fig. Data obtained in experiments carried out in HEK cells in which HKI and HKII were knocked down and in which HKII was overexpressed.**
(XLSX)

**S11 Fig. Data obtained in experiments carried out in HEK cells in which HKI had been overexpressed and which have been incubated for 24hrs in the presence of 6-Aminonicotinamide (6-AN).**
(XLSX)

**S12 Fig. Data in this worksheet were used as a control for the experimental data presented in S11 Fig.** In this case the cells were not incubated in the presence of 6-Aminonicotinamide (6-AN). Experimental data presented in S10 and S11 Figs and obtained with and without 6-AN were recorded on the same day.
(XLSX)

## Author Contributions

**Conceptualization:** Scott John, Guillaume Calmettes, Shili Xu, Bernard Ribalet.

**Formal analysis:** Guillaume Calmettes, Bernard Ribalet.

**Methodology:** Scott John, Shili Xu, Bernard Ribalet.

**Project administration:** Bernard Ribalet.

**Supervision:** Bernard Ribalet.

**Writing – original draft:** Bernard Ribalet.

**Writing – review & editing:** Scott John, Guillaume Calmettes, Shili Xu.

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
