## [Decision Letter · Decision Letter 0]

28 Sep 2022

PONE-D-22-18715Real-time resolution studies of the regulation of pyruvate-dependent lactate metabolism by hexokinases in single cells.PLOS ONE

Dear Dr. Ribalet,

Thank you for submitting your manuscript to PLOS ONE. After careful consideration, we feel that it has merit but does not fully meet PLOS ONE’s publication criteria as it currently stands. Therefore, we invite you to submit a revised version of the manuscript that addresses the points raised during the review process.

We look forward to receiving your revised manuscript.

Kind regards,

Prasanth Puthanveetil

Academic Editor

PLOS ONE

Journal Requirements:

In your cover letter, please note whether your blot/gel image data are in Supporting Information or posted at a public data repository, provide the repository URL if relevant, and provide specific details as to which raw blot/gel images, if any, are not available. Email us at plosone@plos.org if you have any questions

3. Please note that PLOS ONE has specific guidelines on code sharing for submissions in which author-generated code underpins the findings in the manuscript. In these cases, all author-generated code must be made available without restrictions upon publication of the work. Please review our guidelines at https://journals.plos.org/plosone/s/materials-and-software-sharing#loc-sharing-code and ensure that your code is shared in a way that follows best practice and facilitates reproducibility and reuse

5. We note that you have stated that you will provide repository information for your data at acceptance. Should your manuscript be accepted for publication, we will hold it until you provide the relevant accession numbers or DOIs necessary to access your data. If you wish to make changes to your Data Availability statement, please describe these changes in your cover letter and we will update your Data Availability statement to reflect the information you provide

Additional Editor Comments (if provided):

Dear Authors,

Our reviewers have found the manuscript entitled -"Real-time resolution studies of the regulation of pyruvate-dependent lactate metabolism by hexokinases in single cells" have generated some enthusiasm but at the same time there are some concerns that need to be addressed. Kindly respond to our reviewer's concerns appropriately so that it can be further considered.

Will wait to hear from your side.

Best wishes,

Academic Editor

Reviewers' comments:

Reviewer's Responses to Questions

**Comments to the Author**

1. Is the manuscript technically sound, and do the data support the conclusions?

Reviewer #1: Partly

Reviewer #2: Yes

2. Has the statistical analysis been performed appropriately and rigorously? 

Reviewer #1: No

Reviewer #2: Yes

3. Have the authors made all data underlying the findings in their manuscript fully available?

Reviewer #1: No

Reviewer #2: Yes

4. Is the manuscript presented in an intelligible fashion and written in standard English?

Reviewer #1: Yes

Reviewer #2: Yes

5. Review Comments to the Author

Reviewer #1: This manuscript is overall well written and presented and has some findings of interest. There is however, some flaws to be addressed and additional data which is required to fulfill a scientifically sound manuscript for publication.

Main concerns:

-All experiments were conducted in HEK cells. Key experiments must be shown in at least one additional system.

-Representative images of the FRET must be shown in main or supplementary figures to illustrate robustness of signal and system in additional to only the quantified data.

- Figure 1a. Dox label is incomplete, and western bands are not aligned in different panels.

- Many figures do not indicate number of experimental repeats, and need to have statistics and error bars shown (e.g. 4b-d)

Reviewer #2: Article: Real-time resolution studies of the regulation of pyruvate-dependent lactate metabolism by hexokinases in single cells.

Summary –

In this work, the authors have investigated the regulation of lactate metabolism in the light of hexokinase and mitochondrial activity in HEK cells. Through utilization of shRNA mediated knockdown of HKI and HKII coupled with FRET sensor/imaging experiments they have shown that lactate utilization by mitochondria involving lactate to pyruvate conversion in mitochondrial space involves the malate aspartate shuttle and is inhibited by HKs binding in OMM.

Recommendation –

1. Can the authors elaborate on how they are achieving single cell resolution in the FRET sensor assays with laconic? I understand that genetically-encoded reporters will give a higher resolution for the FRET ratios but is it really done at a single cell level? Are there cell sorting or isolation methods which are being used to resolve to single cell basis? I read through the FRET sensor and imaging method space but is not immediately clear. If it is transfected cDNAs in HEK cells then the assay is more heterogenous if the cells are not manipulated/sorted post transfection. Maybe I am missing something here and need more clarification how it is single cell resolution?

2. Just a bit of image alignment here but the labels on Fig1A for -/+ dox are shifted and not aligning with the lanes on the images I received. A graphical normalization of knockdown can also be presented in the supplemental figures.

3. Have the authors investigated ROS activity in the HKI and HKII knockdown cells? This might be useful to related to how lactate utilization can be affect by ROS and also even in states of increased ROS activity if the MA shuttle becomes further de-regulated by HK inhibition along with increase in ROS.

6. PLOS authors have the option to publish the peer review history of their article (what does this mean?). If published, this will include your full peer review and any attached files.

Reviewer #1: No

Reviewer #2: No

---

## [Author Response · Author response to Decision Letter 0]

4 Dec 2022

Response to reviewers:

The authors wish to thank the reviewers for their constructive concerns and comments. We have addressed these comments, and our responses are included in the letter below and in the revised manuscript (yellow highlights). 

General comments: We have carried out a number of additional experiments in other cell types e.g. HeLa and IPS cells to validate our studies in HEK cells. Unfortunately, due to the loss of our knockout HEK cell lines – a failure of our liquid nitrogen storage facility- we do not have access to these cells for further experimentation at this time. It had taken us a long time to generate these cell lines and we are trying to get back some of these reagents, but it has been a very slow process.

Specific comments:

Reviewer #1 concerns:

-All experiments were conducted in HEK cells. Key experiments must be shown in at least one additional system.

New data obtained with HeLa cells are presented in Fig S2. This data shows that the effects of pyruvate and NaCN are similar to those observed in HEK cells. We had previously carried out experiments in IPS cells. The FRET studies utilizing the Laconic probe carried out by B.R. in IPS cells showed that inhibition of mitochondrial metabolism varied as a function of glucose phosphorylation (Nakano et al., 2017). Together these data strengthen our studies carried out in HEK cells.

To test for the effects of inhibitors of the glucose 6 phosphate dehydrogenase (G6PDH) or of the malate aspartate shuttle (MAS) we would have to knock down HKI and HKII in a cell line such as HeLa cells. This would require several months of work for us. Our goal is indeed to demonstrate the validity of our hypotheses. But to this end we propose to knock down Aspartate aminotransferase (GOT1) to further establish the role of MAS in pyruvate-induced lactate metabolism, and knock down G6PDH to establish the role of HKs in regulating this process. These enzymes have already been knocked down in HEK cells and we hope to be able to collaborate with the laboratories that generated these cells lines once our work is published.

Our ultimate goal is to carry out these studies in adult cardiac myocytes (ARVM) and neonatal cardiac myocyte (NRVM), which have very different metabolism, different HKs content and different resistance to ischemia.

-Representative images of the FRET must be shown in main or supplementary figures to illustrate robustness of signal and system in additional to only the quantified data.

We have amended Fig 1 to show images of the change in FRET ratio evoked by the addition of lactate to the media and recorded simultaneously from individual HEK cells. These images underlie the strength of the FRET signals and their reproducibility in uniform response- rates and amplitude. They are but an illustrative example of the approach we have adopted throughout the experimental paradigms investigated.

- Figure 1a. Dox label is incomplete, and western bands are not aligned in different panels.

Amended accordingly. We have also added uncropped image of the gels under supporting information (Fig S6).

- Many figures do not indicate number of experimental repeats, and need to have statistics and error bars shown (e.g. 4b-d)

We have added a number of graphs with error bars and all the relevant information in Fig 2C, Fig 5A&B, Fig S4 and S5. The latter two figures are box plots showing the individual data points.

Reviewer #2: Recommendation –

1. Can the authors elaborate on how they are achieving single cell resolution in the FRET sensor assays with laconic? I understand that genetically-encoded reporters will give a higher resolution for the FRET ratios but is it really done at a single cell level? Are there cell sorting or isolation methods which are being used to resolve to single cell basis? I read through the FRET sensor and imaging method space but is not immediately clear. If it is transfected cDNAs in HEK cells then the assay is more heterogenous if the cells are not manipulated/sorted post transfection. Maybe I am missing something here and need more clarification how it is single cell resolution?

Indeed a couple of our colleagues who initially reviewed our manuscript made the comment that we should further detail the FRET technique that we use. We have inserted images in Fig 1 to better explain our approach. This is also further detailed in “Methods”. Yes, the imaging experiments are carried out on single cells cultured on microscope cover slips. We image between 3 to 6 cells at a time and recordings are done from individual ROI. We make recordings from 30 to 100 cells for each experimental condition. We hope that this addendum will address Reviewer #2 comment. 

2. Just a bit of image alignment here but the labels on Fig1A for -/+ dox are shifted and not aligning with the lanes on the images I received. A graphical normalization of knockdown can also be presented in the supplemental figures. 

Amended accordingly see also Fig S6.for graphical analysis.

3. Have the authors investigated ROS activity in the HKI and HKII knockdown cells? This might be useful to related to how lactate utilization can be affect by ROS and also even in states of increased ROS activity if the MA shuttle becomes further de-regulated by HK inhibition along with increase in ROS.

Thank you for the insightful comment. Indeed, we do have a ROS sensitive fluorescent probe and we had carried out some experiments with it a couple of years ago, but we did not pursue this line of research after we lost our HKI/HKII knockdown cells. Our goal is still to further investigate the role of MAS in the regulation of the metabolism of lactate, but our plan now will be to directly target GOT1. Please see also our answer to reviewer #1 comment #1 and our general comment..

Thank you,

Bernard Ribalet for the authors

---

## [Decision Letter · Decision Letter 1]

25 Jan 2023

PONE-D-22-18715R1

Real-time resolution studies of the regulation of pyruvate-dependent lactate metabolism by hexokinases in single cells.

PLOS ONE

Dear Dr. Ribalet,

Thank you for submitting your manuscript to PLOS ONE. After careful consideration, we have found that our reviewers were not much enthusiastic about this work. After addressing the comments, if you would like to submit as a new submission, you are always welcome to submit. Looking forward to receive your future submissions.

I am sorry that we cannot be more positive on this occasion, but hope that you appreciate the reasons for this decision.

Kind regards,

Prasanth Puthanveetil

Academic Editor

PLOS ONE

Reviewers' comments:

Reviewer's Responses to Questions

**Comments to the Author**

1. If the authors have adequately addressed your comments raised in a previous round of review and you feel that this manuscript is now acceptable for publication, you may indicate that here to bypass the “Comments to the Author” section, enter your conflict of interest statement in the “Confidential to Editor” section, and submit your "Accept" recommendation.

Reviewer #2: All comments have been addressed

Reviewer #3: (No Response)

2. Is the manuscript technically sound, and do the data support the conclusions?

Reviewer #2: Yes

Reviewer #3: Partly

3. Has the statistical analysis been performed appropriately and rigorously? 

Reviewer #2: Yes

Reviewer #3: Yes

4. Have the authors made all data underlying the findings in their manuscript fully available?

Reviewer #2: Yes

Reviewer #3: Yes

5. Is the manuscript presented in an intelligible fashion and written in standard English?

Reviewer #2: Yes

Reviewer #3: Yes

6. Review Comments to the Author

Reviewer #2: Thank you for your response. I truly believe utilizing HK1 and HKII knockdown cells would be a good line of questioning besides the GOT1 line you have mentioned. Best of luck on your future research endeavors.

Reviewer #3: The authors are presumably performing their studies under atmospheric conditions enriched with CO2, hyperbaric conditions, as compared to in vivo. In light of the normoxic oxygen levels ranging from ~0-13%, it is difficult to relate the lactate utilization to what might be actually occurring. This confounding issue should be addressed.

It is indeed unfortunate when there is a loss of a critical reagent such as the knockdown Hek cells. We've all been there. However, it's loss and lack of repeat of findings in another cell line wherein the genes were knocked out means that no one else can repeat the study for independent validation nor has the requirement for the findings to be replicated in another cell line been met.

The graphs are still in need of being cleaned up with proper spacing in titles and “R” values need to be readable.

7. PLOS authors have the option to publish the peer review history of their article (what does this mean?). If published, this will include your full peer review and any attached files.

Reviewer #2: No

Reviewer #3: No

- - - - -

---

## [Author Response · Author response to Decision Letter 1]

4 Mar 2023

Response to the Reviewers comments.

Reviewer #2: Thank you for your response. I truly believe utilizing HK1 and HKII knockdown cells would be a good line of questioning besides the GOT1 line you have mentioned. Best of luck on your future research endeavors.

This sounds positive to us and we thank the reviewer for their review and encouragement..

Reviewer #3: The authors are presumably performing their studies under atmospheric conditions enriched with CO2, hyperbaric conditions, as compared to in vivo. In light of the normoxic oxygen levels ranging from ~0-13%, it is difficult to relate the lactate utilization to what might be actually occurring. This confounding issue should be addressed.

This is confusing to us and colleagues in our Department! What does the reviewer mean by hyperbaric conditions, enriched in CO2? 

First, our experimental conditions are the same as those used by laboratories around the world to study metabolism, or any cellular event for that matter, in cultured cells. These conditions are: room temperature and atmospheric pressure, water saturated, with trace amounts of CO2 close to 0.04% and O2 close to 21%. We cannot figure out where the enriched CO2 and 0-13% O2 come from, at no point in the manuscript did we mention the use of hyperbaric conditions. Under our experimental conditions, with 21% O2 mitochondrial respiration is very efficient and allows for study of lactate utilization. 

If the reviewer is referring to the culture conditions, yes, the CO2 level in the incubator is 5% and the O2 level is close to 16% (21-5=16%), the standard tissue culture conditions. Furthermore, these are the concentrations found in the lung’s alveoli and are therefore the in vivo conditions. Again in this case the pressure inside the incubator is normal.

It is indeed unfortunate when there is a loss of a critical reagent such as the knockdown HEK cells. We've all been there. However, it's loss and lack of repeat of findings in another cell line wherein the genes were knocked out means that no one else can repeat the study for independent validation.

Why is it that no one else can repeat our studies for independent validation? 

• For one thing, almost every laboratory nowadays can knock down genes in cultured cells.

• Second. if someone is willing to spend $4000, the HEK hexokinase knockdown cells are commercially available.

https://www.abcam.com/human-hk2-hexokinase-ii-knockout-hek-293-cell-line-ab269485.html

• And third, as our recent data show, it is possible to replicate the experimental data obtained with hexokinase knockdown cells, by simply removing glucose form the perfusion medium. A predicted result since we have demonstrated that the effect of hexokinase requires the enzyme to be catalytically active and glucose is the only substrate for hexokinase.

 Any interested laboratory can use one or several of these approaches to replicate our findings.

Nor has the requirement for the findings to be replicated in another cell line been met.

This is simply not true. We may not have knockdown hexokinase in another cell line, but we have replicated most of the data presented in the submitted manuscript in a number of cell systems.

• First, we added in supplemental data experimental results that were obtained in Hela cells. These results show that the responses to pyruvate and NaCN, in the absence of glucose, are almost identical to the response obtained in HEK cells wild type and hexokinase knockdown.

• We also quoted in the manuscript that similar results were obtained in IPS cells exposed to solutions with and without glucose.

• In addition, similar responses to pyruvate in the presence and absence of glucose were obtained in neonatal rat ventricular myocytes (NRVM).

All of this indicates that what we observe in HEK cells are properties that are common to a variety of cell types, primary and immortalized.

It is true that we have not knocked down hexokinase in a cell line other than HEK cells, but as we have already outlined, an alternative approach consisting of removing glucose with Hela cells achieves similar results. These observations demonstrate that the changes in lactate metabolism observed in wild type and manipulated HEK cells are common to other cell types.

In view of these observations, knocking down hexokinase in another cell line would not significantly strengthen our hypothesis. Instead, we are aiming at knocking down enzymes of the malate aspartate shuttle (GOT1) and pyruvate carboxylase to validate our model for pyruvate-induced lactate utilization.

 The graphs are still in need of being cleaned up with proper spacing in titles and “R” values need to be readable.

We can space the “R” values so that they are easy to read.

In summary, our experimental work and manuscript preparation indicates the thoroughness and reproducibility of the hypothesis. The comments from Reviewer # 3 are unwarranted especially given the misconceptions regarding our experimental conditions (O2 and CO2 levels), and the lack of reproducibility for data validation. On this basis we respectfully request that the Editor reconsider his decision to decline publication of our work.

---

## [Decision Letter · Decision Letter 2]

4 May 2023

PONE-D-22-18715R2

Real-time resolution studies of the regulation of pyruvate-dependent lactate metabolism by hexokinases in single cells.

PLOS ONE

Dear Dr. Ribalet,

Thank you for submitting your manuscript to PLOS ONE. After careful consideration, we feel that it has merit but does not fully meet PLOS ONE’s publication criteria as it currently stands. Therefore, we invite you to submit a revised version of the manuscript that addresses the points raised during the review process.

Please submit your revised manuscript by **27th May, 2023.** If you will need more time than this to complete your revisions, please reply to this message or contact the journal office at plosone@plos.org. Please include the following items when submitting your revised manuscript:

We look forward to receiving your revised manuscript.

Kind regards,

Bashir Sajo Mienda, PhD

Academic Editor

PLOS ONE

Journal Requirements:

1. Thank you for stating the following financial disclosure:

Additional Editor Comments (if provided):

Reviewers' comments:

Reviewer's Responses to Questions

**Comments to the Author**

1. If the authors have adequately addressed your comments raised in a previous round of review and you feel that this manuscript is now acceptable for publication, you may indicate that here to bypass the “Comments to the Author” section, enter your conflict of interest statement in the “Confidential to Editor” section, and submit your "Accept" recommendation.

Reviewer #4: (No Response)

2. Is the manuscript technically sound, and do the data support the conclusions?

Reviewer #4: Yes

3. Has the statistical analysis been performed appropriately and rigorously? 

Reviewer #4: Yes

4. Have the authors made all data underlying the findings in their manuscript fully available?

Reviewer #4: Yes

5. Is the manuscript presented in an intelligible fashion and written in standard English?

Reviewer #4: Yes

6. Review Comments to the Author

Reviewer #4: In this interesting study, John et al provide evidence that hexokinase acts, via pentose phosphate pathway-inhibition of the malate-aspartate shuttle, to inhibit pyruvate-dependent mitochondrial lactate oxidation. This was accomplished by manipulating the expression of hexokinases in HEK 293 cells, and then monitoring lactate in real-time using the FRET probe, laconic. Inhibitors specific to the malate-aspartate shuttle, the pentose phosphate pathway and pyruvate carboxylase, as well as the electron transport chain, were used to test aspects of the overarching scheme. Among the major findings are that in HK knock-down, added pyruvate resulted in decreased lactate. This pyruvate-dependent lactate oxidation required both malate-aspartate shuttle activity and a functioning respiratory chain. HK overexpression attenuated this pyruvate-dependent removal of lactate. Using an inhibitor of the pentose phosphate pathway, the authors conclude that HK acts through the PPP to inhibit MAS, and prevent pyruvate-dependent lactate removal. This study adds to the potential roles of HKs in the regulation of pyruvate and lactate metabolism. The results are discussed within the context of the related (patho)physiology literature.

The manuscript is well written. The study appears to be technically sound, supported with verification experiments, and appropriate statistical analyses. The use of FRET-expression-based probes to monitor lactate in real-time in the intact cell is a strength of the study, as is the varied conditions tested (inhibitors, manipulation of HK expression).

Questions for consideration

1. How might glutamate (and/or malate) compare to pyruvate with respect to lactate removal? In other words, is the pyruvate-dependent lactate oxidation observed in the study a unique function of pyruvate, or can another substrate accomplish the same? Such a comparison would potentially tease out whether PDH activation vs MAS activation are responsible for the pyruvate-dependent decline in lactate observed in the study.

2. In this study, pyruvate carboxylase is considered and tested. How might malic enzyme contribute to the proposed model?

7. PLOS authors have the option to publish the peer review history of their article (what does this mean?). If published, this will include your full peer review and any attached files.

Reviewer #4: No

---

## [Author Response · Author response to Decision Letter 2]

11 May 2023

Guillaume Calmettes received partial salary from the American Heart Association in 2017-2018.

Shili Xu received his salary from a US Dept of Defense grant in 2017-2018

The author(s) did not received any other specific funding for this work.

---

## [Editor Report · Decision Letter 3]

22 May 2023

Real-time resolution studies of the regulation of pyruvate-dependent lactate metabolism by hexokinases in single cells.

PONE-D-22-18715R3

Dear Dr. Ribalet,

We’re pleased to inform you that your manuscript has been judged scientifically suitable for publication and will be formally accepted for publication once it meets all outstanding technical requirements.

Kind regards,

Bashir Sajo Mienda, PhD

Academic Editor

PLOS ONE
---

## [Editor Report · Acceptance letter]

25 May 2023

PONE-D-22-18715R3 

Real-time resolution studies of the regulation of pyruvate-dependent lactate metabolism by hexokinases in single cells. 

Dear Dr. Ribalet:

I'm pleased to inform you that your manuscript has been deemed suitable for publication in PLOS ONE. Congratulations! Your manuscript is now with our production department. 

Kind regards, 

on behalf of

Dr. Bashir Sajo Mienda 

Academic Editor

PLOS ONE